# Diverse viral proteases activate the NLRP1 inflammasome

**Brian V Tsu[1†], Christopher Beierschmitt[1†], Andrew P Ryan[1], Rimjhim Agarwal[2], Patrick S Mitchell[2,3], Matthew D Daugherty[1]\***

[1]Division of Biological Sciences, University of California San Diego, San Diego, United States; [2]Division of Immunology & Pathogenesis, University of California Berkeley, Berkeley, United States; [3]Department of Microbiology, University of Washington, Seattle, United States

**Abstract** The NLRP1 inflammasome is a multiprotein complex that is a potent activator of inflammation. Mouse NLRP1B can be activated through proteolytic cleavage by the bacterial Lethal Toxin (LeTx) protease, resulting in degradation of the N-terminal domains of NLRP1B and liberation of the bioactive C-terminal domain, which includes the caspase activation and recruitment domain (CARD). However, natural pathogen-derived effectors that can activate human NLRP1 have remained unknown. Here, we use an evolutionary model to identify several proteases from diverse picornaviruses that cleave human NLRP1 within a rapidly evolving region of the protein, leading to host-specific and virus-specific activation of the NLRP1 inflammasome. Our work demonstrates that NLRP1 acts as a 'tripwire' to recognize the enzymatic function of a wide range of viral proteases and suggests that host mimicry of viral polyprotein cleavage sites can be an evolutionary strategy to activate a robust inflammatory immune response.

**\*For correspondence:**
mddaugherty@ucsd.edu

[†]These authors contributed equally to this work

**Competing interests:** The authors declare that no competing interests exist.

## Introduction

The ability to sense and respond to pathogens is central to the mammalian immune system. However, immune activation needs to be properly calibrated, as an overactive immune response can at times be as pathogenic as the pathogen itself. To ensure accurate discrimination of self and non-self, innate immune sensors detect broadly conserved microbial molecules such as bacterial flagellin or double-stranded RNA (*Janeway, 1989*). However, such microbial patterns can be found on harmless and pathogenic microbes alike. More recently, pathogen-specific activities such as toxins or effector enzymes have also been shown to be targets of innate immune recognition (*Jones et al., 2016*; *Mitchell et al., 2019*; *Vance et al., 2009*). Such a system for detection, termed effector-triggered immunity (ETI), has been well-established in plants (*Cui et al., 2015*; *Jones et al., 2016*) and is emerging as an important means to allow the immune system to distinguish pathogens from harmless microbes in animals (*Fischer et al., 2020*; *Jones et al., 2016*).

Complicating the success of host detection systems, innate immune sensors are under constant selective pressure to adapt due to pathogen evasion or antagonism of immune detection. Such evolutionary dynamics, termed host-pathogen arms races, result from genetic conflicts where both host and pathogen are continually driven to adapt to maintain a fitness advantage. The antagonistic nature of these conflicts can be distinguished via signatures of rapid molecular evolution at the exact sites where host and pathogen interact (*Daugherty and Malik, 2012*; *Meyerson and Sawyer, 2011*; *Sironi et al., 2015*). Consistent with their role as the first line of cellular defense against incoming pathogens, innate immune sensors of both broad molecular patterns as well as specific pathogen-associated effectors have been shown to be engaged in genetic conflicts with pathogens (*Cagliani et al., 2014*; *Chavarría-Smith et al., 2016*; *Hancks et al., 2015*; *Tenthorey et al., 2014*; *Tian et al., 2009*).

**eLife digest** The immune system recognizes disease-causing microbes, such as bacteria and viruses, and removes them from the body before they can cause harm. When the immune system first detects these foreign invaders, a multi-part structure known as the inflammasome launches an inflammatory response to help fight the microbes off. Several sensor proteins can activate the inflammasome, including one in mice called NLRP1B. This protein has evolved a specialized site that can be cut by a bacterial toxin. Once cleaved, this region acts like a biological tripwire and sparks NLRP1B into action, allowing the sensor to activate the inflammasome system.

Humans have a similar protein called NLRP1, but it is unclear whether this protein has also evolved a tripwire region that can sense microbial proteins. To answer this question, Tsu, Beierschmitt et al. set out to find whether NLRP1 can be activated by viruses in the *Picornaviridae* family, which are responsible for diseases like polio, hepatitis A, and the common cold. This revealed that NLRP1 contains a cleavage site for enzymes produced by some, but not all, of the viruses in the picornavirus family. Further experiments confirmed that when a picornavirus enzyme cuts through this region during a viral infection, it triggers NLRP1 to activate the inflammasome and initiate an immune response.

The enzymes from different viruses were also found to cleave human NLRP1 at different sites, and the protein's susceptibility to cleavage varied between different animal species. For instance, Tsu, Beierschmitt et al. discovered that NLRP1B in mice is also able to sense picornaviruses, and that different enzymes activate and cleave NLRP1B and NLRP1 to varying degrees: this affected how well the two proteins are expected to be able to sense specific viral infections. This variation suggests that there is an ongoing evolutionary arms-race between viral proteins and the immune system: as viral proteins change and new ones emerge, NLRP1 rapidly evolves new tripwire sites that allow it to sense the infection and launch an inflammatory response.

What happens when NLRP1B activates the inflammasome during a viral infection is still an open question. The discovery that mouse NLRP1B shares features with human NLRP1 could allow the development of animal models to study the role of the tripwire in antiviral defenses and the overactive inflammation associated with some viral infections. Understanding the types of viruses that activate the NLRP1 inflammasome, and the outcomes of the resulting immune response, may have implications for future treatments of viral infections.

Inflammasomes are one such group of rapidly evolving cytosolic immune sensor complexes (*Broz and Dixit, 2016*; *Chavarría-Smith et al., 2016*; *Evavold and Kagan, 2019*; *Rathinam and Fitzgerald, 2016*; *Tenthorey et al., 2014*; *Tian et al., 2009*). Upon detection of microbial molecules or pathogen-encoded activities, inflammasome-forming sensor proteins serve as a platform for the recruitment and activation of proinflammatory caspases including caspase-1 (CASP1) through either a pyrin domain (PYD) or a caspase activation and recruitment domain (CARD) (*Broz and Dixit, 2016*; *Rathinam and Fitzgerald, 2016*). Active CASP1 mediates the maturation and release of the proinflammatory cytokines interleukin (IL)−1β and IL-18 (*Broz and Dixit, 2016*; *Rathinam et al., 2012*). CASP1 also initiates a form of cell death known as pyroptosis (*Broz and Dixit, 2016*; *Rathinam et al., 2012*). Together, these outputs provide robust defense against a wide array of eukaryotic, bacterial, and viral pathogens (*Broz and Dixit, 2016*; *Evavold and Kagan, 2019*; *Rathinam and Fitzgerald, 2016*).

The first described inflammasome is scaffolded by the sensor protein NLRP1, a member of the nucleotide-binding domain (NBD), leucine-rich repeat (LRR)-containing (NLR) superfamily (*Martinon et al., 2002*; *Ting et al., 2008*). NLRP1 has an unusual domain architecture, containing a CARD at its C-terminus rather than the N-terminus like all other inflammasome sensor NLRs, and a function-to-find domain (FIIND), which is located between the LRRs and CARD (*Ting et al., 2008*). The FIIND undergoes a constitutive self-cleavage event, such that NLRP1 exists in its non-activated state as two, noncovalently associated polypeptides (*D'Osualdo et al., 2011*; *Finger et al., 2012*; *Frew et al., 2012*), the N-terminal domains and the C-terminal CARD-containing fragment.

The importance of the unusual domain architecture of NLRP1 for mounting a pathogen-specific inflammasome response has been elucidated over the last several decades (*Evavold and Kagan,*

*2019*; *Mitchell et al., 2019*; *Taabazuing et al., 2020*). The first hint that NLRP1 does not detect broadly conserved microbial molecules came from the discovery that the *Bacillus anthracis* Lethal Toxin (LeTx) is required to elicit a protective inflammatory response against *B. anthracis* infection via one of the mouse NLRP1 homologs, NLRP1B (*Boyden and Dietrich, 2006*; *Greaney et al., 2020*; *Moayeri et al., 2010*; *Terra et al., 2010*). Paradoxically, inflammasome activation is the result of site-specific cleavage in the N-terminus of mouse NLRP1B by the Lethal Factor (LF) protease subunit of LeTx, indicating that protease-mediated cleavage of NLRP1 does not disable its function but instead results in its activation (*Chavarría-Smith and Vance, 2013*; *Levinsohn et al., 2012*). More recently, the mechanism by which LF-mediated proteolytic cleavage results in direct NLRP1B inflammasome activation has been detailed (*Chui et al., 2019*; *Sandstrom et al., 2019*). These studies revealed that proteolysis of mouse NLRP1B by LF results in exposure of a novel N-terminus, which is then targeted for proteasomal degradation by a protein quality control mechanism called the 'N-end rule' pathway (*Chui et al., 2019*; *Sandstrom et al., 2019*; *Wickliffe et al., 2008*; *Xu et al., 2019*). Since the proteasome is a processive protease, it progressively degrades the N-terminal domains of NLRP1B but is disengaged upon arriving at the self-cleavage site within the FIIND domain. Degradation of the N-terminal domains thus releases the bioactive C-terminal CARD-containing fragment of NLRP1B from its non-covalent association with the N-terminal domains, which is sufficient to initiate downstream inflammasome activation (*Chui et al., 2019*; *Sandstrom et al., 2019*). By directly coupling NLRP1 inflammasome activation to cleavage by a pathogen-encoded protease, NLRP1 can directly sense and respond to the activity of a pathogen effector. Such a model indicates that the N-terminal domains are not required for NLRP1 activation per se, but rather serve a pathogen-sensing function. Interestingly, the N-terminal 'linker' region in mouse NLRP1B that is cleaved by LF is rapidly evolving in rodents, and the analogous linker region is likewise rapidly evolving in primate species (*Chavarría-Smith et al., 2016*). These data suggest that selection from pathogens has been driving diversification of this protease target region of NLRP1, possibly serving to bait diverse pathogenic proteases into cleaving NLRP1 and activating the inflammasome responses.

Consistent with the rapid evolution in NLRP1 at the site of proteolytic cleavage, LF neither cleaves nor activates human NLRP1 (*Mitchell et al., 2019*; *Moayeri et al., 2012*; *Taabazuing et al., 2020*). Despite this, human NLRP1 can also be activated by proteolysis when a tobacco etch virus (TEV) protease site is engineered into the rapidly evolving linker region of human NLRP1 that is analogous to the site of LF cleavage in mouse NLRP1B (*Chavarría-Smith et al., 2016*). Thus, like mouse NLRP1B, it has been predicted that human NLRP1 may serve as a 'tripwire' sensor for pathogen-encoded proteases (*Mitchell et al., 2019*).

Here, we investigate the hypothesis that viral proteases cleave and activate human NLRP1. We reasoned that human viruses, many of which encode proteases as necessary enzymes for their replication cycle, may be triggers for NLRP1 activation. To pursue this hypothesis, we focused on viruses in the *Picornaviridae* family, which encompass a diverse set of human enteric and respiratory pathogens including coxsackieviruses, polioviruses, and rhinoviruses (*Zell, 2018*). These viruses all translate their genome as a single polyprotein, which is then cleaved into mature proteins in at least six sites in a sequence-specific manner by a virally encoded 3C protease, termed 3C$^{pro}$ (*Laitinen et al., 2016*; *Solomon et al., 2010*; *Sun et al., 2016*; *Zell, 2018*). 3C$^{pro}$ is also known to proteolytically target numerous host factors, many of which are associated with immune modulation (*Croft et al., 2018*; *Huang et al., 2015*; *Lei et al., 2017*; *Mukherjee et al., 2011*; *Qian et al., 2017*; *Wang et al., 2019*; *Wang et al., 2012*; *Wang et al., 2014*; *Wang et al., 2015*; *Wen et al., 2019*; *Xiang et al., 2014*; *Xiang et al., 2016*; *Zaragoza et al., 2006*). Because 3C$^{pro}$ are evolutionarily constrained to cleave several specific polyprotein sites and host targets for replicative success, we reasoned that human NLRP1 may have evolved to sense viral infection by mimicking viral polyprotein cleavage sites, leading to NLRP1 cleavage and inflammasome activation. Using an evolution-guided approach, we now show that NLRP1 is cleaved in its rapidly evolving linker region by several 3C$^{pro}$ from picornaviruses, resulting in inflammasome activation. These results are consistent with the recent discovery that human rhinovirus (HRV) 3C$^{pro}$ cleaves and activates NLRP1 in airway epithelia (*Robinson et al., 2020*). We also find that variation in the cleavage site among primates, and even within the human population, leads to changes in cleavage susceptibility and inflammasome activation. Interestingly, we observe that proteases from multiple genera of viruses cleave and activate human NLRP1 and mouse NLRP1B at different sites, supporting a role for an evolutionary conflict between viral proteases and NLRP1. Taken together, our work highlights the role of NLRP1 in

sensing and responding to diverse viral proteases by evolving cleavage motifs that mimic natural sites of proteolytic cleavage in the viral polyprotein.

## Results

### Human NLRP1 contains mimics of viral protease cleavage sites

Our hypothesis that NLRP1 can sense viral proteases is based on two prior observations. First, both human NLRP1 and mouse NLRP1B can be activated by N-terminal proteolysis (*Chavarría-Smith et al., 2016*). Second, the linker in primate NLRP1, which is analogous to the N-terminal disordered region of NLRP1B that is cleaved by LF protease, has undergone recurrent positive selection (*Chavarría-Smith et al., 2016*), or an excess of non-synonymous amino acid substitutions over what would be expected by neutral evolution (*Kimura, 1983*). We reasoned that this rapid protein sequence evolution may reflect a history of pathogen-driven selection, wherein primate NLRP1 has evolved to sense pathogen-encoded proteases such as those encoded by picornaviruses. To test this hypothesis, we first generated a predictive model for 3C$^{pro}$ cleavage site specificity. We chose to focus on the enterovirus genus of picornaviruses, as there is a deep and diverse collection of publicly available viral sequences within this genus due to their importance as human pathogens including coxsackieviruses, polioviruses, enterovirus D68, and HRV (*Blom et al., 1996*; *Pickett et al., 2012*). We first compiled complete enterovirus polyprotein sequences from the Viral Pathogen Resource (ViPR) database (*Pickett et al., 2012*) and extracted and concatenated sequences for each of the cleavage sites within the polyproteins (*Figure 1A and B*, *Supplementary files 1* and *2*). After removing redundant sequences, we used the MEME Suite (*Bailey et al., 2009*) to create the following 3C$^{pro}$ cleavage motif: [A/Φ]XXQGXXX (where Φ denotes a hydrophobic residue and X denotes any amino acid), which is broadly consistent with previous studies that have experimentally profiled the substrate specificity of enterovirus 3C$^{pros}$ (*Blom et al., 1996*; *Fan et al., 2020*; *Jagdeo et al., 2018*; *O'Donoghue et al., 2012*; *Figure 1C*).

We next optimized our 3C$^{pro}$ cleavage site motif prediction by querying against predicted viral polyprotein and experimentally validated host cleavage sites (*Laitinen et al., 2016*), allowing us to set thresholds for predicting new cleavage sites (*Supplementary files 3* and *4*). Due to the low-information content of the polyprotein motif (*Figure 1C*), such predictions are necessarily a compromise between stringency and capturing the most known cleavage sites. In particular, we wished to make sure that the model was able to capture a majority of experimentally validated human hits (compiled in *Laitinen et al., 2016*) in addition to the known sites of polyprotein cleavage ('true positives'), while minimizing the prediction of sites outside of known polyprotein cleavage sites ('false positives'). By adjusting the model to allow greater flexibility for amino acids not sampled in the viral polyprotein (see Materials and methods and *Figure 1—figure supplement 1* and *Supplementary file 4*), we were able to capture 95% of known viral sites and the majority of the known human hits, while limiting the number of false negative hits within the viral polyprotein (*Figure 1D*).

### The coxsackievirus B3 3C$^{pro}$ cleaves human NLRP1 at a predicted site within the linker region

We next used our refined model to conduct a motif search for 3C$^{pro}$ cleavage sites in NLRP1 using Find Individual Motif Occurrences (FIMO) (*Grant et al., 2011*). We identified three occurrences of the motif across the full-length human NLRP1 protein (*Figure 2A*). Of these sites, one in particular, 127-GCTQGSER-134, fell within the previously described rapidly evolving linker (*Chavarría-Smith et al., 2016*) and demonstrates the lowest percent conservation across mammalian species at each of the predicted P4-P4' positions (*Figure 2B*).

To assess if human NLRP1 is cleaved by enteroviral 3C$^{pro}$, we co-expressed a N-terminal mCherry-tagged wild-type (WT) human NLRP1 with the 3C$^{pro}$ from the model enterovirus, coxsackievirus B3 (CVB3) in HEK293T cells (*Figure 2C*). The mCherry tag stabilizes and allows visualization of putative N-terminal cleavage products, similar to prior studies (*Chavarría-Smith et al., 2016*). We observed that the WT but not catalytically inactive (C147A) CVB3 3C$^{pro}$ cleaved NLRP1, resulting in a cleavage product with a molecular weight consistent with our predicted 3C$^{pro}$ cleavage at the predicted 127-GCTQGSER-134 site (44 kDa) (*Figure 2D*). Based on the presence of a single cleavage product, we assume that the other predicted sites are either poor substrates for 3C$^{pro}$ or less

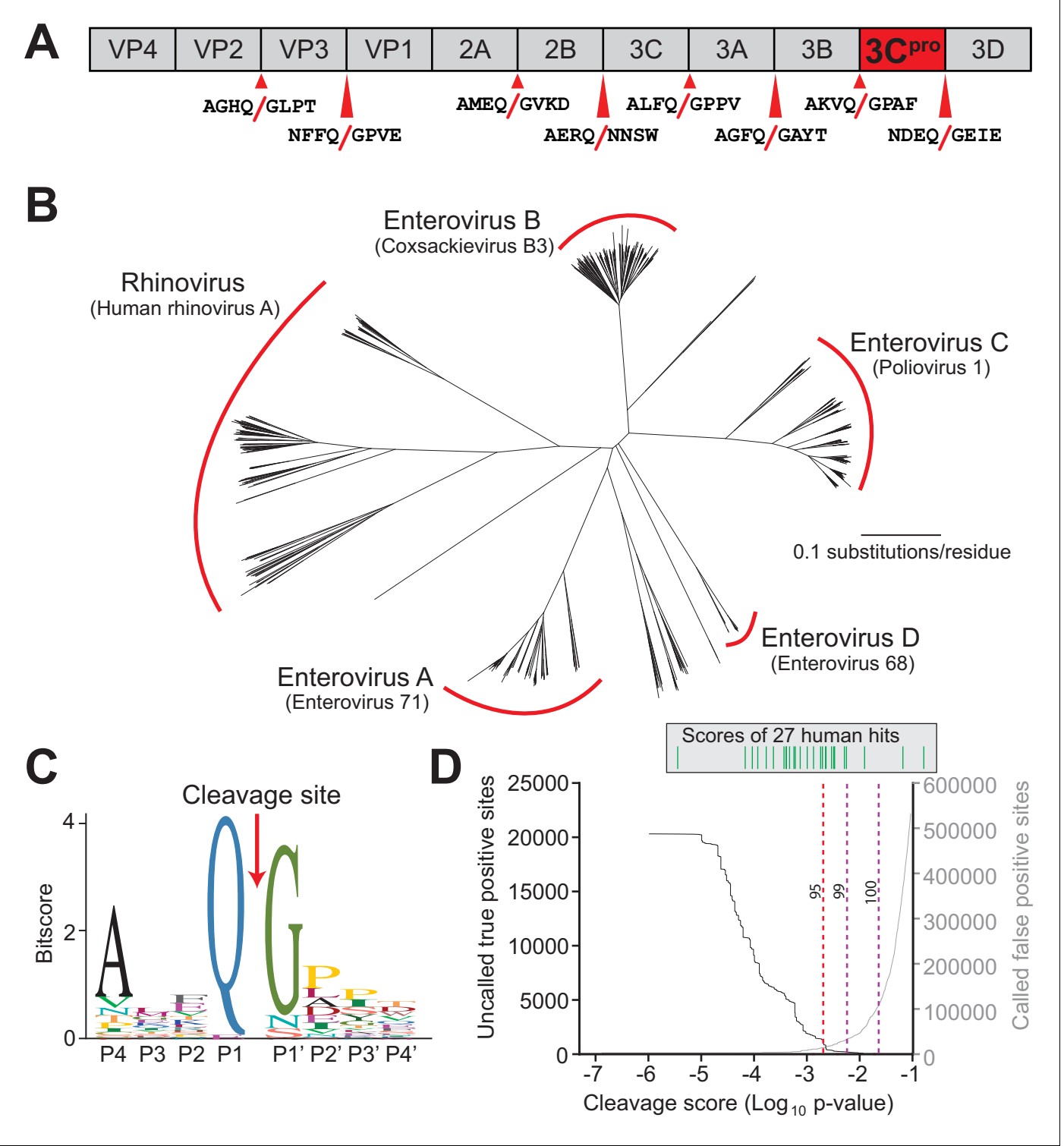

**Figure 1.** Conserved polyprotein cleavage sites across enteroviruses inform substrate specificity of the enteroviral 3C[pro]. (A) Schematic of 3C[pro] cleavage sites (red arrows) within the polyprotein of coxsackievirus B3 Nancy (CVB3), a model enterovirus. Shown are the eight amino acids flanking each cleavage site within the polyprotein. (B) Phylogenetic tree of 796 enteroviral polyprotein coding sequences depicting the major clades of enteroviruses sampled in this study with representative viruses from each clade in parentheses (*Supplementary file 2*). (C) Eight amino acid polyprotein cleavage motif for enteroviruses (labeled as positions P4 to P4') generated from the 796 enteroviral polyprotein sequences in (B) using the MEME Suite (*Supplementary file 2*). (D) Training set data used to determine the motif search threshold for FIMO (*Supplementary files 1*, *3* and *4*). The X-axis represents a $\log_{10}$ of the p-value reported by FIMO as an indicator for the strength of the cleavage motif hit (cleavage score). (Left) The Y-axis depicts

*Figure 1 continued on next page*

*Figure 1 continued*

the number of uncalled true positives, or motif hits that overlap with the initial set of 8mer polyprotein cleavage sites used to generate the motif, in the training set of enteroviral polyprotein sequences (black). (Right) The Y-axis depicts the number of called false positive sites, or any motif hits found in the polyprotein that are not known to be cleaved by 3C$^{pro}$, in the training set of enteroviral polyprotein sequences (gray). (Above) Each line depicts a single, experimentally validated case of enteroviral 3C$^{pro}$ cleavage site within a human protein as reported in *Laitinen et al., 2016* and is ordered along the X-axis by its resulting cleavage score. A vertical dotted line is used to represent the decided threshold that captures 95% of true positive hits and 16 out of 27 reported human hits (*Figure 1—figure supplement 1*).

The online version of this article includes the following figure supplement(s) for figure 1:

**Figure supplement 1.** Motif optimization enhances capture of known human targets of enteroviral 3C$^{pro}$.

accessible to the protease as would be predicted from their NetSurfP-reported (*Klausen et al., 2019*) coil probability within structured domains of the protein (*Figure 2A* and *Figure 2—source data 1*). To determine if the cleavage occurs between residues 130 and 131, we mutated the P1' glycine to a proline (G131P), which abolished 3C$^{pro}$ cleavage of NLRP1 (*Figure 2D*). CVB3 3C$^{pro}$ cleavage of NLRP1 resulted in a similarly intense cleavage product when compared to the previously described system in which a TEV protease site was introduced into the linker region of NLRP1 (*Chavarría-Smith et al., 2016*; *Figure 2D*). Taken together, these results indicate that cleavage of WT NLRP1 by a protease from a natural human pathogen is robust and specific.

During a viral infection, 3C$^{pro}$ is generated in the host cell cytoplasm after translation of the viral mRNA to the polyprotein and subsequent processing of the viral polyprotein into constituent pieces (*Laitinen et al., 2016*). To confirm that virally-produced 3C$^{pro}$, or the 3 CD precursor that can also carry out proteolytic cleavage during infection (*Laitinen et al., 2016*), is able to cleave NLRP1, we virally infected cells expressing either WT NLRP1 or the uncleavable (G131P) mutant. We observed accumulation of the expected cleavage product beginning at 6 hr post-infection when we infected cells expressing WT NLRP1 and no cleavage product when we infected cells expressing the 131P mutant (*Figure 2E*). These results validate that CVB3 infection can result in rapid and specific cleavage of human NLRP1.

## The CVB3 3C$^{pro}$ activates human NLRP1 by cleaving within the linker region

Previous results with a TEV-cleavable human NLRP1 showed that cleavage by TEV protease was sufficient to activate the human NLRP1 inflammasome in a reconstituted inflammasome assay (*Chavarría-Smith et al., 2016*). Using the same assay, in which plasmids-encoding human NLRP1, CASP1, ASC, and IL-1β are transfected into HEK293T cells, we tested if the CVB3 3C$^{pro}$ activates the NLRP1 inflammasome. We observed that the CVB3 3C$^{pro}$ results in robust NLRP1 inflammasome activation, as measured by CASP1-dependent processing of pro-IL-1β to the active p17 form (*Figure 3A*). As expected, CVB3 3C$^{pro}$ activation of the NLRP1 inflammasome was prevented by introduction of a mutation in the NLRP1 FIIND (S1213A) (*D'Osualdo et al., 2011*; *Finger et al., 2012*; *Frew et al., 2012*; *Figure 3—figure supplement 1* – panel A), which prevents FIIND auto-processing and the release of the bioactive C-terminal UPA–CARD (*Chui et al., 2019*; *Sandstrom et al., 2019*). Consistent with recent results (*Robinson et al., 2020*), we also observed that chemical inhibitors of the proteasome (MG132) or the Cullin-RING E3 ubiquitin ligases that are required for the degradation of proteins with a novel N-terminal glycine (MLN4924) (*Timms et al., 2019*), also blocked CVB3 3C$^{pro}$ activation of NLRP1 (*Figure 3—figure supplement 1* – panel B). To confirm that 3C$^{pro}$-induced inflammasome activation resulted in release of bioactive IL-1β from cells, we measured active IL-1β levels in the culture supernatant using cells engineered to express a reporter gene in response to soluble, active IL-1β. When compared to a standard curve (*Figure 3—figure supplement 2*), we found that 3C$^{pro}$ treatment resulted in release of >4 ng/ml of active IL-1β into the culture supernatant (*Figure 3B*). Importantly, in both western blot and cell culture assays, 3C$^{pro}$-induced inflammasome activation was comparable to TEV-induced activation and was ablated when position 131 was mutated, validating that CVB3 3C$^{pro}$ cleavage at a single site is both necessary and sufficient to activate NLRP1 (*Figure 3A and B*). Taken together, our results are consistent with CVB3 3C$^{pro}$ activating the NLRP1 inflammasome via site-specific cleavage and subsequent 'functional degradation' (*Chui et al., 2019*; *Sandstrom et al., 2019*).

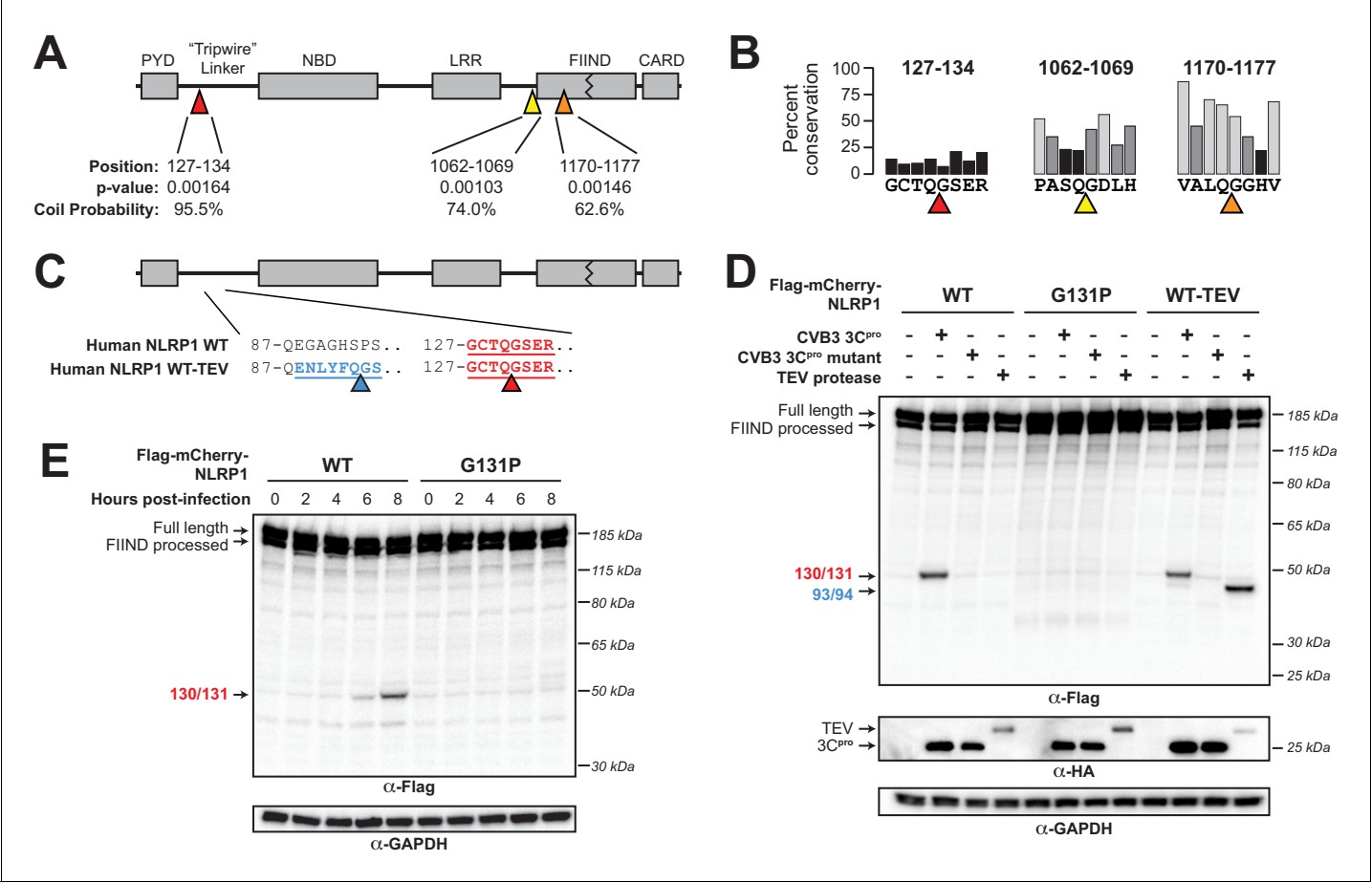

**Figure 2.** Enterovirus 3C^pro cleaves human NLRP1 at the predicted site of mimicry. (A) Schematic of the domain structure of NLRP1, with predicted cleavage sites (triangles). FIMO-reported p-values and average NetsurfP-reported coil probabilities (*Figure 2—source data 1*) are described at the predicted sites. (B) Percent conservation across 100 mammalian species at each position of each predicted 8mer cleavage site within human NLRP1. (C) Schematic of the human NLRP1 sequence used to assess enteroviral cleavage and activation. The predicted enteroviral cleavage site found in the linker region (127-GCTQGSER-134) is shown in red. Human NLRP1 WT-TEV contains an engineered TEV cleavage site between residues 93 and 94 (underlined green) in human NLRP1 WT. (D) Immunoblot depicting human NLRP1 cleavage by CVB3 3C^pro and TEV protease. HEK293T cells were co-transfected using 100 ng of the indicated Flag-tagged mCherry-NLRP1 fusion plasmid constructs with 250 ng of the indicated protease construct and immunoblotted with the indicated antibodies. (E) Immunoblot depicting human NLRP1 cleavage at the indicated timepoints after infection with 250,000 PFU (MOI = ~1) CVB3. HEK293T cells were transfected using 100 ng of either WT NLRP1 or NLRP1 G131P and infected 24–30 hr later. All samples were harvested 32 hr post-transfection and immunoblotted with the indicated antibodies.

The online version of this article includes the following source data for figure 2:

**Source data 1.** Tabular output of NetSurfP structural predictions for human NLRP1.

We next wished to test whether CVB3 infection, through the site-specific cleavage of NLRP1 by 3C^pro, is able to activate the NLRP1 inflammasome. Consistent with our prediction, recent work has revealed that HRV infection can cleave and activate human NLRP1 in airway epithelia (*Robinson et al., 2020*). However, prior work has also implicated a role for the NLRP3 inflammasome in enterovirus infection (*Kuriakose and Kanneganti, 2019*; *Xiao et al., 2019*), including activation of the NLRP3 inflammasome during CVB3 infection in mice and human cell lines (*Wang et al., 2019*; *Wang et al., 2018*). NLRP1 and NLRP3 have distinct expression patterns (*Robinson et al., 2020*; *Zhong et al., 2016*) including in epithelial cells, which are important targets of enterovirus infection. NLRP3 is activated in response to various noxious stimuli or damage signals associated with pathogen infection (*Evavold and Kagan, 2019*; *Spel and Martinon, 2021*). In contrast, NLRP1 is activated by direct proteolytic cleavage of its N-terminal 'tripwire' region by viral proteases. We therefore wished to confirm that specific 3C^pro cleavage of NLRP1 during CVB3 infection is able to activate the NLRP1 inflammasome. We first virally infected 293 T cells, which do not express either NLRP1 or

NLRP3, that were co-transfected with either WT NLRP1 or the uncleavable (G131P) mutant in our reconstituted inflammasome assay and measured active IL-1β in the culture supernatant. Eight hours after infection with CVB3, we observe robust release of active IL-1β into the culture supernatant when cells were transfected with WT NLRP1 but not the uncleavable mutant NLRP1 (*Figure 3C*). To test whether CVB3 infection can activate the inflammasome in an NLRP1-dependent fashion in cells that naturally express an intact NLRP1 inflammasome, we took advantage of the fact that NLRP1 has been described as the primary inflammasome in human keratinocytes (*Zhong et al., 2016*). We therefore infected WT, *NLRP1*, or *CASP1* KO (*Figure 3—figure supplement 3*) immortalized HaCaT

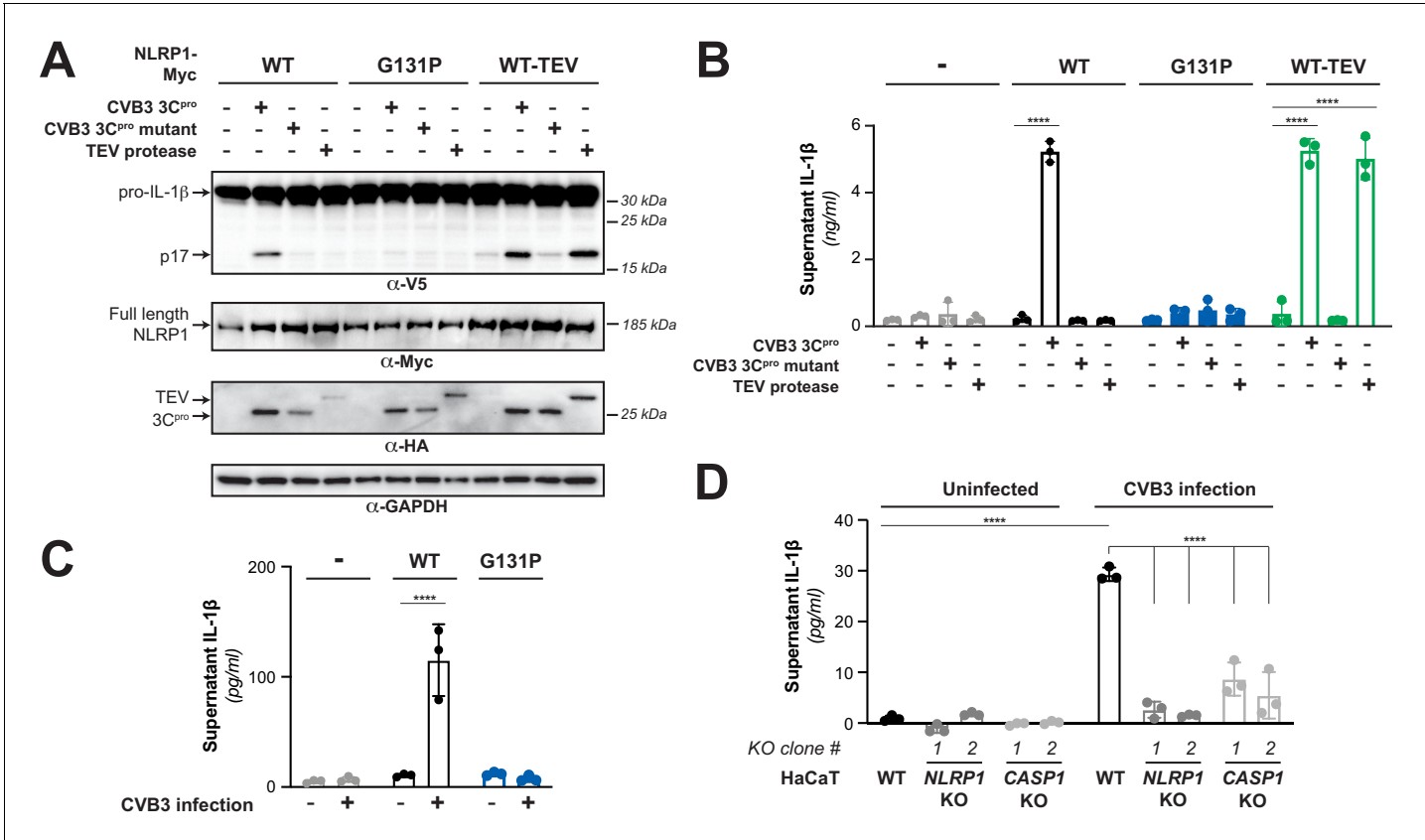

**Figure 3.** Enterovirus 3C[pro] cleavage of human NLRP1 promotes pro-inflammatory cytokine release. (A) Immunoblot depicting human NLRP1 activation (maturation of IL-1β) by CVB3 3C[pro] and TEV protease. HEK293T cells were co-transfected using 100 ng of the indicated protease, 50 ng V5-IL-1β, 100 ng CASP1, 5 ng ASC, and 4 ng of the indicated Myc-tagged NLRP1, and immunoblotted with the indicated antibodies. Appearance of the mature p17 band of IL-1β indicates successful assembly of the NLRP1 inflammasome and activation of CASP1. (B) Bioactive IL-1β in the culture supernatant was measured using HEK-Blue IL-1β reporter cells, which express secreted embryonic alkaline phosphatase (SEAP) in response to extracellular IL-1β. Supernatant from cells transfected as in (A) was added to HEK-Blue IL-1β reporter cells and SEAP levels in the culture supernatant from HEK-Blue IL-1β reporter cells were quantified by the QUANTI-Blue colorimetric substrate. Transfections were performed in triplicate and compared to the standard curve generated from concurrent treatment of HEK-Blue IL-1β reporter cells with purified human IL-1β (*Figure 3—figure supplement 2*). Data were analyzed using two-way ANOVA with Sidak's post-test. **** = p<0.0001. (C) CVB3 infection of inflammasome-reconstituted HEK293T cells results in IL-1β release when NLRP1 can be cleaved by 3C[pro]. Cells were transfected with the indicated NLRP1 construct and other NLRP1 inflammasome components as in (B). Sixteen hours post-transfection, cells were mock infected or infected with 250,000 PFU (MOI = ~1) CVB3. Eight hours post-infection, culture supernatant was collected and bioactive IL-1β was measured as in (B). (D) CVB3 infection of an immortalized human keratinocyte cell line, HaCaT, activates the NLRP1 inflammasome. WT or knockout (*Figure 3—figure supplement 3*) HaCaT cell lines were mock infected or infected with 100,000 PFU (MOI = ~0.4) CVB3. Forty-eight hours post-infection, culture supernatant was collected and bioactive IL-1β was measured as in (B). The online version of this article includes the following source data and figure supplement(s) for figure 3:

**Source data 1.** Individual data values for *Figure 3B, C and D*.

**Figure supplement 1.** 3C[pro]-mediated activation of the human NLRP1 inflammasome depends on FIIND autoprocessing and proteosomal degradation.

**Figure supplement 2.** Standard curve for *Figure 3B*.

**Figure supplement 3.** Validation of CRISPR/Cas9-editing of *NLRP1* or *CASP1* in HaCaT cells by Sanger sequencing.

human keratinocytes with CVB3 and measured release of active IL-1β in the culture supernatant. Consistent with our model that CVB3 infection cleaves and activates the NLRP1 inflammasome, we observe a significant increase in supernatant IL-1β after CVB3 infection that is reduced in cells that lack either NLRP1 or CASP1 (*Figure 3D*). Together, these results indicate that CVB3 infection, through 3C$^{pro}$ cleavage of the tripwire region of NLRP1, activates the NLRP1 inflammasome.

## NLRP1 diversification across primates and within humans confers host differences in susceptibility to viral 3C$^{pro}$ cleavage and inflammasome activation

Our evolutionary model in which NLRP1 is evolving in conflict with 3C$^{pro}$ suggests that changes in the NLRP1 linker region, both among primates and within the human population (*Figure 4A*), would confer host-specific differences to NLRP1 cleavage and inflammasome activation. To test this hypothesis, we aligned the linker regions from NLRP1 from diverse mammals and human population sampling and compared the sequences around the site of CVB3 3C$^{pro}$ cleavage (*Figure 4B and C* and *Figure 4—figure supplement 1*). We noted that while a majority of primate NLRP1s are predicted to be cleaved similarly to the human ortholog, several primate proteins would be predicted to not be cleaved by enteroviral 3C$^{pro}$ as a result of changes to either the P4, P1 or P1' residues. To confirm these predictions, we made the human NLRP1 mutants G127E or G131R, which reflect the Old World monkey or marmoset residues at each position, respectively. As predicted, both primate NLRP1 variants prevented 3C$^{pro}$ cleavage of NLRP1 (*Figure 4D*). These results indicate that multiple viral 3C$^{pro}$ activate host NLRP1 in a host specific manner and suggest that single changes within a short linear motif can substantially alter cleavage susceptibility and inflammasome activation.

We further observed that this cleavage site is largely absent in non-primate species (*Figure 4—figure supplement 1*), suggesting that a 3C$^{pro}$ cleavage site mimic emerged in simian primates 30–40 million years ago. While many other mammalian species have a region that is alignable to the primate linker, we noted that this region is unalignable to any sequence in the linker region of NLRP1 proteins from rodents or bats (*Figure 4B* and *Figure 4—figure supplement 1*). Despite this, we found that there was weak cleavage of mouse NLRP1B at a site closer to the N-terminus than the 127-GCTQGSER-134 site found in human NLRP1 (*Figure 4D* and Figure 6A), suggesting that an independent cleavage site could have arisen elsewhere in mouse NLRP1B. These data suggest that NLRP1 in other mammals may have convergently evolved cleavage sites in the linker region despite not having a cleavable sequence in the precise position that human NLRP1 is cleaved.

Differential host susceptibility to NLRP1 cleavage and activation extends to the human population level. Using GnomAD (*Karczewski et al., 2020*), we sampled the alternative alleles within the direct cleavage site (*Figure 4C*). While this region does not appear to be highly polymorphic in humans, we note that one alternative allele (rs150929926) results in a Q130R mutation and is present in >1 in every 1000 African alleles sampled. Introducing this mutation into NLRP1, we find the Q130R mutation eliminates NLRP1 cleavage susceptibility to CVB3 3C$^{pro}$ (*Figure 4E*). In the case of primate and human diversity alleles at the site of 3C$^{pro}$ cleavage, we also find that loss of cleavage susceptibility results in a loss of inflammasome activation in response to 3C$^{pro}$ (*Figure 4F*), supporting the aforementioned notion that single changes in the linker region can have drastic impacts on the ability of different hosts to respond to the presence of cytoplasmic 3C$^{pro}$.

## 3C$^{pro}$ from diverse picornaviruses cleave and activate human NLRP1

Our evolutionary model predicted that NLRP1 would be cleaved by a broad range of 3C$^{pro}$ from viruses in the enterovirus genus (*Figure 1B*). To test this hypothesis, we cloned 3C$^{pro}$ from representative viruses from four additional major species of human enteroviruses: enterovirus 71 (EV71, species: *Enterovirus A*), poliovirus 1 (PV1, species: *Enterovirus C*), enterovirus D68 (EV68, species: *Enterovirus D*), human rhinovirus A (HRV, species: *Rhinovirus A*), in order to compare them to the 3C$^{pro}$ from CVB3 (species: *Enterovirus B*) (*Figure 5A*). Despite <50% amino acid identity between some of these proteases (*Figure 5—figure supplement 1*; *Figure 5—figure supplement 1—source data 1*), the overall structures of these proteases are similar (*Figure 5—figure supplement 2*) and the cleavage motifs are closely related (*Figure 5A*). Consistent with this predicted target similarity and prior data with HRV (*Robinson et al., 2020*), we found that every tested member of enterovirus 3C$^{pro}$ was able to cleave NLRP1 between residues 130 and 131 (*Figure 5B*). Moreover, expression

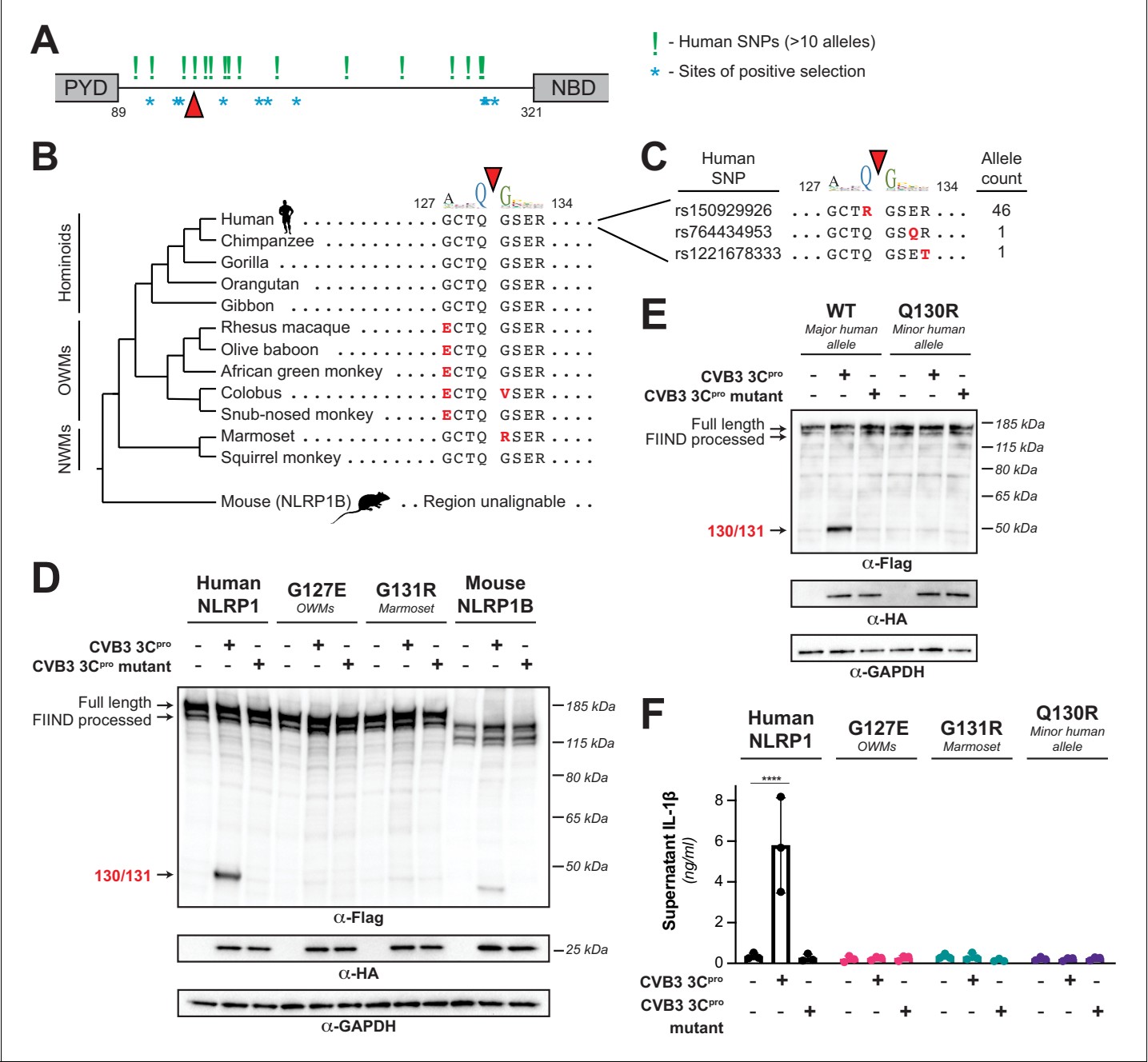

**Figure 4.** Naturally occurring cleavage site variants alter NLRP1 susceptibility to enteroviral 3C$^{pro}$. (A) Schematic of sites found to be evolving under positive selection (marked as *, from *Chavarría-Smith and Vance, 2013*) and human SNPs with at least 10 reported instances in the Genome Aggregation Database (GnomAD, *Karczewski et al., 2020*) (marked as !) within the linker region between the pyrin domain (PYD) and nucleotide binding domain (NBD) of NLRP1. The enteroviral 3C$^{pro}$ cleavage site between position 130 and 131 is indicated by a red triangle. (B) Phylogenetic tree depicting the enteroviral 3C$^{pro}$ cleavage site (red triangle) within NLRP1 across three clades of primates – hominoids, Old World monkeys (OWMs), and New World monkeys (NWMs). Mouse NLRP1B lacks any sequence that is alignable to this region of primate NLRP1 (see also *Figure 4—figure supplement 1*). Amino acid differences to the human NLRP1 reference sequence are highlighted in red. Above the alignment is the enterovirus 3C$^{pro}$ sequence logo shown in *Figure 1*. (C) GnomAD-derived allele counts of each missense human SNP (by reference SNP #) within the 8mer of the determined enteroviral 3C$^{pro}$ cleavage site. (D–E) Immunoblot depicting CVB3 3C$^{pro}$ cleavage susceptibility of the indicated 8mer site variants introduced into human NLRP1 or full-length wild-type mouse NLRP1B (129 allele) (D) or the cleavage susceptibility of human NLRP1 Q130R, a naturally occurring human population variant (E). (F) Release of bioactive IL-1β into the culture supernatant was measured using HEK-Blue IL-1β reporter cells as in *Figure 3B*. Data were analyzed using two-way ANOVA with Sidak's post-test. **** = p<0.0001.

The online version of this article includes the following source data and figure supplement(s) for figure 4:

*Figure 4 continued on next page*

*Figure 4 continued*

**Source data 1.** Individual data values for *Figure 4F*.
**Figure supplement 1.** Mammalian NLRP1 phylogenomics and alignment of linker region.

of every tested enterovirus 3C^pro resulted in activation of the inflammasome in a manner that was dependent on cleavage at the 127-GCTQGSER-134 site (*Figure 5C*).

Enteroviruses are only one genus within the broad *Picornaviridae* family of viruses. We next asked if viruses in other *Picornaviridae* genera that infect humans are also able to cleave and activate human NLRP1. We were unable to generate a robust sequence motif for every genera of picornavirus due to lower depth of publicly available sequences. Instead, we cloned a 3C^pro from a representative of every genus of picornavirus that are known to infect humans: encephalomyocarditis virus (EMCV, genus: *Cardiovirus*), parechovirus A virus (ParA, genus: *Parechovirus*), Aichi virus (Aichi, genus: *Kobuvirus*), hepatitis A virus (HepA, genus: *Hepatovirus*), salivirus A virus (SaliA, genus: *Salivirus*), and rosavirus A2 (Rosa2, genus: *Rosavirus*). Each of these viral proteases is <20% identical to CVB3 3C^pro (*Figure 5—figure supplement 1—source data 1*). Despite this, the sequence motif built from cleavage sites within the polyprotein of these individual viruses is broadly consistent with the motif seen in enteroviruses (*Figure 5A*), reflective of the strong evolutionary constraint on evolution of the sequence specificity of these proteases and overall structural conservation of the active sites of these proteases (*Figure 5—figure supplement 2*). Interestingly, we found that there was substantial variation in NLRP1 cleavage sites across these diverse 3C^pro even though most picornavirus proteases cleaved human NLRP1 to some degree (*Figure 5B*). For instance, while 3C^pro from EMCV and ParA did not cleave NLRP1, we observed distinct cleavage sites for 3C^pro from Aichi, HepA, SaliA and Rosa2 (*Figure 5B*), all of which have at least one cleavage site predicted to occur in the linker region (expected size between 40 kDa and 67 kDa). Confirming that these proteases cleave at a site that is distinct from that of enteroviruses, the G131P NLRP1 mutant is still cleaved by the non-enteroviral proteases (*Figure 5B*).

Surprisingly, when we interrogated NLRP1 inflammasome activation by 3C^pros from Aichi, HepA, SaliA, and Rosa2, all of which robustly cleave NLRP1 at a site in the linker region, we found that only Rosa2 was able to activate the NLRP1 inflammasome (*Figure 5C*). While it is possible that NLRP1 cleavage by 3C^pro from these other viruses is too weak or in a region that may be inconsistent with activation, we also noted that there are obvious cleavage sites in NLRP1 that are outside of the linker region and closer to the FIIND autocleavage site. Cleavage at these sites in NLRP1, or cleavage of other host genes, may interfere with activation that may have otherwise been induced by 3C^pro cleavage in the linker region. Indeed, we find that co-expression of 3C^pro from Aichi, HepA, SaliA can attenuate NLRP1 activation by TEV protease (*Figure 5—figure supplement 3*), consistent with the idea that these three proteases can actively block NLRP1 activation. Further investigation will be needed to determine the exact mechanism by which this occurs. Nevertheless, our data demonstrate that non-enteroviral 3C^pros can cleave NLRP1 at independent sites in the rapidly evolving linker region and can, in at least one case, activate the human NLRP1 inflammasome.

To further confirm that 3C^pro cleavage (or lack thereof) of NLRP1 is reflective of 3C^pro during viral infection, we infected cells expressing WT or 131P NLRP1 with EMCV. Consistent with our co-transfection experiments, we see no cleavage of NLRP1 when we infect with EMCV, despite seeing robust cleavage when we infect with CVB3 (*Figure 5D*). Likewise, we see no IL-1β release when we infect either inflammasome-reconstituted HEK293T cells or inflammasome-competent HaCaT cells with EMCV (*Figure 5—figure supplement 4*). These data indicate that evolution of viral 3C^pro cleavage specificity alters whether a virus can be sensed by the NLRP1 tripwire or not.

## Enterovirus 3C^pro cleaves and activates mouse NLRP1B in a virus- and host allele-specific manner

Two bacterial pathogen effectors are known to activate mouse NLRP1B, the LF protease from *B. anthracis* (*Boyden and Dietrich, 2006*; *Greaney et al., 2020*; *Moayeri et al., 2010*; *Terra et al., 2010*); (*Chavarría-Smith and Vance, 2013*; *Levinsohn et al., 2012*) and the IpaH7.8 E3 ubiquitin ligase from *Shigella flexneri* (*Sandstrom et al., 2019*). Interestingly, in both of these cases, activation is specific to the 129 allele of mouse NLRP1B, whereas the B6 allele of NLRP1B is not activated by

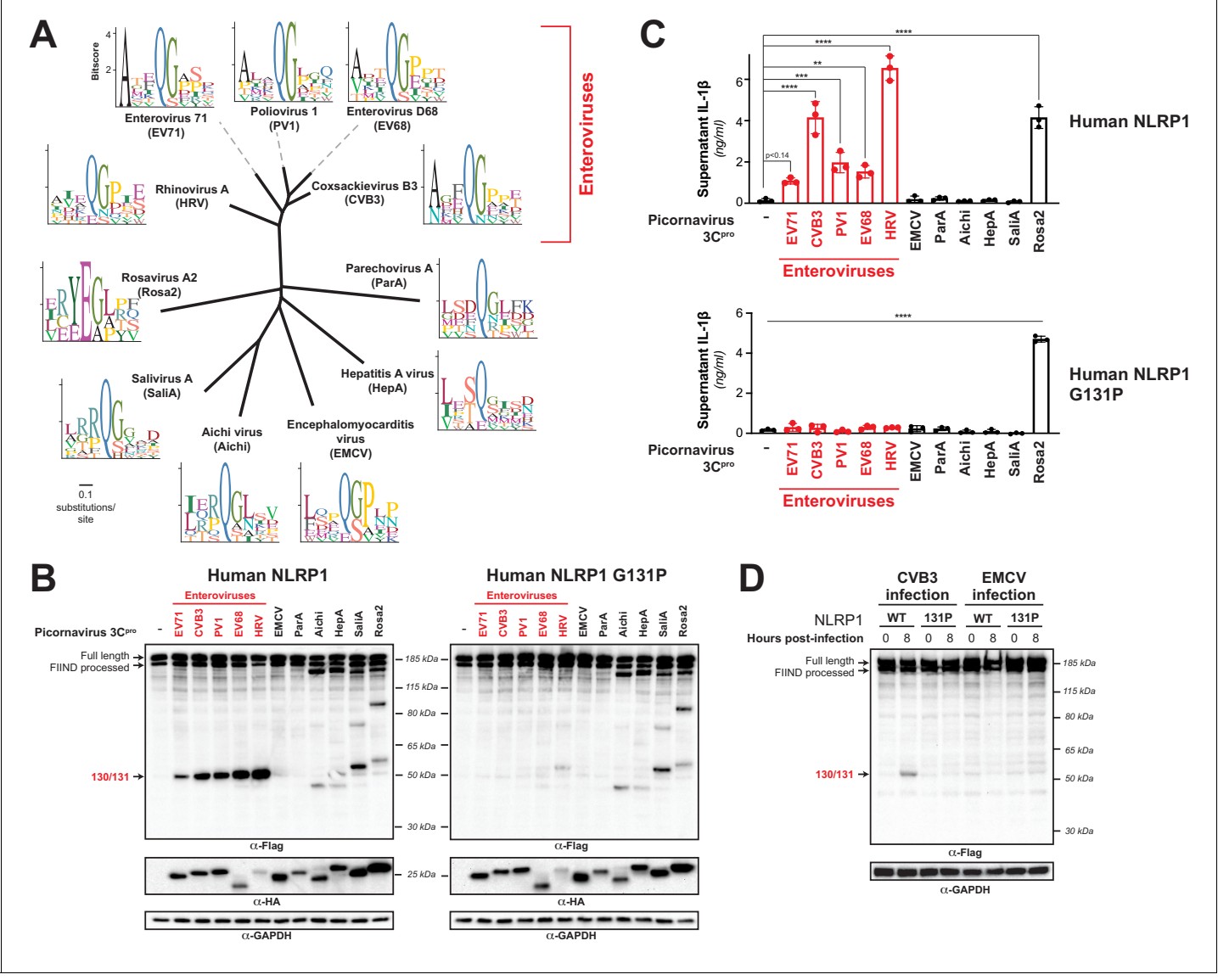

**Figure 5.** Diverse picornavirus 3C^pros cleave and activate NLRP1 at independently evolved sites. (A) Phylogenetic tree of 3C^pro protein sequences for the indicated picornaviruses (*Figure 5—figure supplement 1*; *Figure 5—figure supplement 1—source data 1*). Shown next to the virus name is the sequence motif generated from the known sites of 3C^pro polyprotein cleavage in that specific virus. (B) Immunoblot depicting human NLRP1 cleavage by the indicated picornaviral 3C^pro. Abbreviations are as in (A). Assays were performed as in *Figure 2D*. (left) Cleavage assays against WT NLRP1. (right) Human NLRP1 G131P mutant used in *Figure 2*. (C) Release of bioactive IL-1β into the culture supernatant was measured using HEK-Blue IL-1β reporter cells as in *Figure 3B*. Data were analyzed using one-way ANOVA with Tukey's post-test. ** = p<0.01, *** = p<0.001, **** = p<0.0001. (D) Immunoblot depicting human NLRP1 cleavage at the indicated timepoints after infection with 250,000 PFU (MOI = ~1) CVB3 or EMCV. HEK293T cells were transfected using 100 ng of either WT NLRP1 or NLRP1 G131P and, 24 hr later, either mock infected (0 hr timepoint) or infected with CVB3 or EMCV as indicated (8 hr timepoint). All samples were harvested 32 hr post-transfection and immunoblotted with the indicated antibodies.

The online version of this article includes the following source data and figure supplement(s) for figure 5:

**Source data 1.** Individual data values for *Figure 5C* and *Figure 5—figure supplements 3* and *4*.

**Figure supplement 1.** Alignment of 3C^pros used in this study.

**Figure supplement 1—source data 1.** Table of pairwise percent sequence identity of 3C^pros used in this study as determined from the alignment shown in *Figure 5—figure supplement 1*.

**Figure supplement 2.** Structural similarity of picornavirus 3C^pros.

**Figure supplement 3.** Inhibition of NLRP1 activation by non-enteroviral 3C^pro.

**Figure supplement 4.** EMCV infection does not activate the NLRP1 inflammasome.

these pathogenic effectors. Given the power of mouse models for understanding inflammasome biology, we wished to determine if 3C$^{pros}$ cleave and activate mouse NLRP1B.

Strikingly, when we co-transfected NLRP1B from either the 129 or the B6 strains with diverse enterovirus 3C$^{pros}$, we observed allele-specific cleavage products (*Figure 6A*). Consistent with data in *Figure 4D*, we observed weak cleavage of 129 NLRP1B by CVB3 3C$^{pro}$. In addition, we found that 3C$^{pro}$ from other enteroviruses varied substantially in their ability to cleave 129 NLRP1B, including no detectable cleavage with EV71 3C$^{pro}$ and a different dominant position of cleavage by HRV 3C$^{pro}$. Despite this variation, we only observed weak cleavage (*Figure 6A*, left) and little to no inflammasome activation (*Figure 6B*, left) by any enterovirus 3C$^{pros}$ tested against 129 NLRP1B. In contrast, enterovirus 3C$^{pro}$ cleavage of B6 NLRP1B resulted in a consistent-sized cleavage product

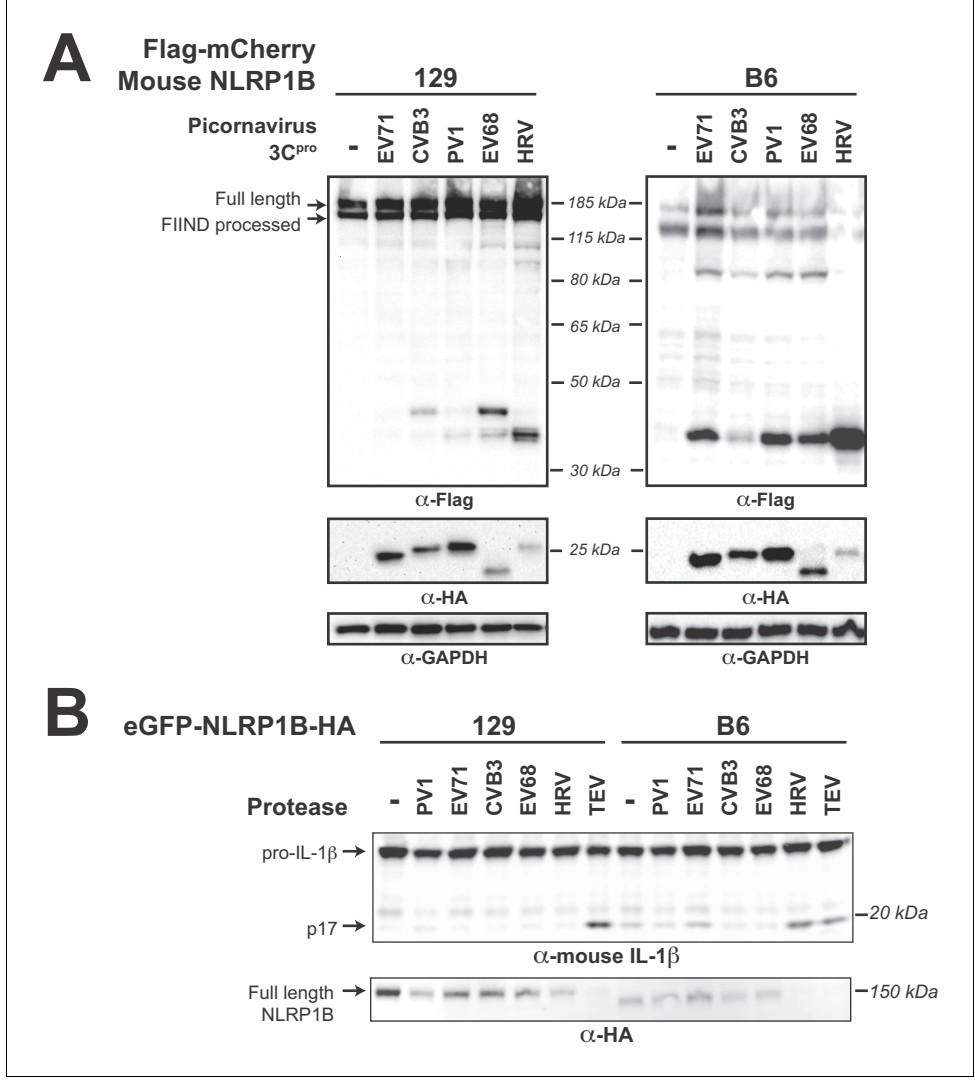

**Figure 6.** Diverse picornavirus 3C$^{pros}$ cleave and activate mouse NLRP1B at independently evolved sites. (**A**) Immunoblot depicting CVB3 3C$^{pro}$ cleavage susceptibility of two alleles (129 and B6) of mouse NLRP1B. Assays were performed as in *Figure 2D*. (**B**) Immunoblot depicting mouse NLRP1B activation (maturation of IL-1β) by enterovirus 3C$^{pro}$ and TEV protease. HEK293T cells were co-transfected using 100 ng of the indicated protease, 50 ng mouse IL-1β, 50 ng mouse CASP1, and either 4 ng of 129 NLRP1B or 2.5 ng of B6 NLRP1B constructs, and immunoblotted with the indicated antibodies. Appearance of the mature p17 band of IL-1β indicates successful assembly of the NLRP1B inflammasome and activation of CASP1.

The online version of this article includes the following figure supplement(s) for figure 6:

**Figure supplement 1.** Alignment of N-termini of mouse NLRP1B 129 and B6 alleles.

across all enterovirus 3C^pros that ranged in intensities between the different viral proteases, more similar to our observations with human NLRP1. Most interestingly, we observed that co-transfection with HRV 3C^pro resulted in the appearance of a very strong cleavage product (*Figure 6A*, right), almost complete loss of full length B6 NLRP1B (*Figure 6A and B*, right) and very strong activation of the inflammasome (*Figure 6B*, right). These data indicate that mouse NLRP1B can also be cleaved and activated by viral proteases, which suggests that the evolution of the N-terminus of NLRP1B between closely related mouse strains (*Figure 6—figure supplement 1*) is not only shaping susceptibility to tripwire cleavage by the bacterial LF protease, but also impacts tripwire cleavage by viral 3C^pros. Taken together, these data further support the model in which both host and viral evolution, even within closely related host and viral species, shape the outcome of the interaction between NLRP1 and 3C^pro.

## Discussion

Pathogens and their hosts are locked in a continual evolutionary conflict in which each side is attempting to exploit the others' weakness. One particularly successful strategy that pathogens have adopted is to exploit host processes that are highly constrained, leaving the host little room to evolutionarily adapt to overcome the pathogen. For instance, molecular mimicry of host proteins is commonly deployed by pathogens to antagonize host defenses, as it limits the evolutionary options for the host to counter-evolve (*Elde and Malik, 2009*). Beyond mimicry of entire proteins or protein domains, pathogens can also mimic so-called 'short linear motifs' (SLIMs) through evolution of only a small number of amino acids to hijack highly conserved host processes such as post-translational modifications or binding by small protein domains (*Chemes et al., 2015*; *Hagai et al., 2014*). Although these strategies are generally described as taking advantage of host evolutionary constraint, pathogens also have potential weak points of evolutionary constraint. In particular, proteases from positive-sense RNA viruses, such as picornaviruses, need to specifically cleave numerous sites within the viral polyprotein in order to reproduce. Thus, changing protease specificity requires concomitant changes to several independent cleavage sites, which is difficult to accomplish in a single evolutionary step. On top of that, protease cleavage motifs often only span a small number of amino acids (*Schechter and Berger, 1967*), potentially facilitating the independent evolution of these SLIMs in host proteins.

Here, we show that the inflammasome protein, NLRP1, serves as a sensor for diverse proteases from the *Picornaviridae* family of human pathogens by mimicking the highly conserved protease cleavage sites found within the viral polyproteins. By exploiting a constrained feature of viral evolution and tying it to a pro-inflammatory immune response, such a system allows the immune system to recognize and respond to a wide range of viral proteases expressed in the host cytoplasm. NLRP1 represents one of the few known cases of mammalian ETI (*Cui et al., 2015*; *Fischer et al., 2020*; *Jones et al., 2016*), where pathogen-mediated cleavage of NLRP1 promotes its activation. By holding the small C-terminal CARD-containing fragment in a non-covalent association with the larger N-terminal fragment, the majority of the protein can serve as a sensor for pathogen-encoded effectors (*Mitchell et al., 2019*; *Taabazuing et al., 2020*). This presents an opportunity to allow NLRP1 to evolve to be recognized by pathogenic effectors, ultimately leading to degradation of the N-terminal fragment. Indeed, mouse NLRP1B has been shown to be specifically cleaved by the protease-containing secreted effector from *B. anthracis* (LF) as well as being ubiquitylated by an E3-ubiquitin ligase from *S. flexneri* (IpaH7.8) (*Sandstrom et al., 2019*). While these two examples provide evidence that the mouse NLRP1B inflammasome operates by a 'functional degradation' model, a direct pathogen-encoded activator of human NLRP1 had remained elusive. We now show, using an evolution-guided approach, that proteases from diverse picornaviruses, including human pathogens such as coxsackievirus B3 (CVB3), human rhinovirus A (HRV), enterovirus D68 (EV68) and poliovirus 1 (PV1) and rosavirus A2 (Rosa2), specifically cleave several independently evolved sites in human NLRP1, leading to activation of the NLRP1 inflammasome and release of pro-inflammatory cytokines such as IL-1β. Together with recent findings (*Robinson et al., 2020*), our work has thus identified proteases from a diverse range of picornaviruses as pathogen-encoded activators of human NLRP1.

We previously speculated that the unique domain architecture of NLRP1 would allow the N-terminal linker of human NLRP1 to freely evolve to be recognized by pathogenic effectors. Indeed, by harvesting publicly available enterovirus polyprotein sequences for known 3C^pro cleavage sites, we

created a 3C$^{pro}$ cleavage motif that was used to successfully predict the site of enterovirus 3C$^{pro}$ cleavage at position 130–131 within the rapidly-evolving linker NLRP1. Additionally, our finding that numerous enteroviruses also cleave at the Q130-G131 site and activate pro-inflammatory cytokine release suggests that human NLRP1 serves as a general enteroviral protease sensor by encoding a polyprotein cleavage site mimic. Our phylogenetic assessment of the Q130-G131 3C$^{pro}$ cleavage site in NLRP1 suggests that NLRP1 sensing of enteroviruses at this specific site is an innovation in the primate lineage, and is largely absent in all other mammalian lineages with exception of a possible independent acquisition by members within the *Caprinae* subfamily of mammals (e.g. goats, sheep) (*Figure 3—figure supplement 1*). Interestingly, even within the primate lineage and a small fraction of the human population, some primate orthologs and human variants are cleavage-resistant and therefore do not activate the inflammasome upon cytoplasmic expression of 3C$^{pro}$. Such data may hint at three different possible explanations for these changes. First, evolutionary drift in the absence of pressure from pathogenic enteroviruses may account for loss of enterovirus 3C$^{pro}$ responsiveness in these genes. Second, selection to sense another viral protease may shape the same region of the linker. Finally, while the ETI model of NLRP1 suggests that enteroviral cleavage of NLRP1 has evolved to activate a beneficial immune response in certain contexts, the effects of NLRP1 overactivation may be detrimental in other contexts. In human skin keratinocytes, where NLRP1 is regarded as the key inflammasome, all components of the NLRP1 inflammasome are basally expressed and thus poised to elicit an inflammatory response (*Zhong et al., 2016*). Here, germline mutations in NLRP1 that result in overactivation can cause growth of warts in the upper airway in a condition known as recurrent respiratory papillomatosis (JRRP) (*Drutman et al., 2019*) and an increase in skin cancer susceptibility and skin disorders such as multiple self-healing palmoplantar carcinoma (MSPC), familial keratosis lichenoides chronica (FKLC) and auto-inflammation with arthritis and dyskeratosis (AIADK) (*Grandemange et al., 2017*; *Herlin et al., 2020*; *Soler et al., 2013*; *Zhong et al., 2016*; *Zhong et al., 2018*). Additional recent work has indicated that dsRNA can also activate the NLRP1 inflammasome in human keratinocytes (*Bauernfried et al., 2020*), adding to the role that NLRP1 may play in the inflammatory response. Beyond the skin, NLRP1 is also basally expressed in tissues such as the gut and brain (*D'Osualdo et al., 2015*; *Kaushal et al., 2015*; *Kummer et al., 2007*), which are sites of picornavirus replication where overactivation upon infection may result in immunopathology. Further in vivo studies will help determine the role of NLRP1 in antiviral immunity and/or immunopathology during viral infection. Facilitating these studies, our discovery that 3C$^{pro}$ from HRV potently cleaves and activates NLRP1B from B6 but not 129 mice suggests that rhinovirus infection of B6 mice may be a good model for studying the in vivo consequences of viral-mediated NLRP1 inflammasome activation.

Intriguingly, 3C$^{pros}$ from nearly every genus of human-infecting picornavirus can cleave NLRP1 somewhere in the rapidly evolving linker region between the PYD and NLR domain, although only enteroviruses cleave at the specific site between position 130 and 131. These data suggest that this extended linker, which we previously found showed widespread signatures of positive selection (*Chavarría-Smith et al., 2016*), may be convergently evolving to mimic cleavage sites from a diverse range of viruses at multiple independent sites. Supporting that model, we observe a similar phenomenon in mouse NLRP1B, where multiple viral proteases cleave at different sites within NLRP1 in a strain-specific manner. These data highlight the important functional differences in cleavage specificity between even closely related 3C$^{pro}$ that are not accounted for by predictive models. Further studies will be required to understand the precise relationships between sites within NLRP1 and individual protease specificity. Intriguingly, not all these cleavage events lead to inflammasome activation in the same way that enteroviral cleavage does, and we find evidence for antagonism of NLRP1 activation by some 3C$^{pros}$, suggesting that additional activities of 3C$^{pro}$ may be the next step in the arms race, serving to prevent inflammasome activation even after the tripwire has been tripped.

Taken together, our work suggests that host mimicry of viral polyprotein cleavage motifs could be an important evolutionary strategy in the ongoing arms race between host and viruses. Indeed, one explanation for the somewhat surprising observation that the specificity of viral proteases changes at all within a viral family such as the picornaviruses is that there is evolutionary pressure from the host to evolve cleavage sites and protease specificity. Prior work has highlighted the roles that viral proteases can play in antagonizing host immune factors and driving host evolution to avoid being cleaved (*Patel et al., 2012*; *Stabell et al., 2018*). In that case, the viral proteases would evolve

to antagonize new factors while maintaining polyprotein cleavage. However, mimicry coupled with cleavage-activating immunity as seen with NLRP1 could be an even stronger pressure to shape the protease specificity. By turning the tables, these host processes may drive the type of functional diversification of viral protease specificity that we observe in order to avoid cleaving NLRP1 and other similar ETI factors. We expect that this work may lead to the discovery that such an evolutionary strategy may be more broadly deployed at other sites of host-pathogen conflicts.

# Materials and methods

## Key resources table

| Reagent type (species) or resource | Designation | Source or reference | Identifiers | Additional information |
|---|---|---|---|---|
| Gene (*Homo sapiens*) | NLRP1 | NCBI | NCBI: NP_127497.1 | |
| Gene (*Mus musculus*) | NLRP1B (129) | NCBI | NCBI: AAZ40510.1 | |
| Gene (*Mus musculus*) | NLRP1B (B6) | NCBI | NCBI: XM_017314698.2 | |
| Cell line (*Homo sapiens*) | HEK293T | ATCC | Cat# CRL-3216; RRID:CVCL_0063 | |
| Cell line (*Homo sapiens*) | HEK-Blue IL-1β cells | Invivogen | HKB-IL1B | |
| Cell line (*Homo sapiens*) | HaCaT (parental) | UC Berkeley Cell Culture Facility | | |
| Cell line (*Homo sapiens*) | HaCaT Cas9 (WT) | This paper | | |
| Cell line (*Homo sapiens*) | HaCaT Cas9 ΔNLRP1 #1 (NLRP1 KO clone #1) | This paper | | Exon five target (TCCACTGCTTGTACGAGACT) |
| Cell line (*Homo sapiens*) | HaCaT Cas9 ΔNLRP1 #2 (NLRP1 KO clone #2) | This paper | | Exon two target (TGTAGGGGAATGAGGGAGAG) |
| Cell line (*Homo sapiens*) | HaCaT Cas9 ΔCASP1 #1 (CASP1 KO clone #1) | This paper | | Exon two target (CCAAACAGACAAGGTCCTGA) |
| Cell line (*Homo sapiens*) | HaCaT Cas9 ΔCASP1 #2 (CASP1 KO clone #2) | This paper | | Exon two target (CCAAACAGACAAGGTCCTGA) |
| Recombinant DNA reagent | pcDNA5/FRT/TO (plasmid) | Invitrogen | V652020 | |
| Recombinant DNA reagent | pQCXIP (plasmid) | Takara Bio | 631516 | |
| Recombinant DNA reagent | psPAX2 (plasmid) | Addgene | RRID:Addgene_12260 | Gift from Dr. Didier Trono |
| Recombinant DNA reagent | pMD2.G (plasmid) | Addgene | RRID:Addgene_12259 | Gift from Dr. Didier Trono |
| Recombinant DNA reagent | pLB-Cas9 (plasmid) | Addgene | RRID:Addgene_52962 | Gift from Dr. Feng Zhang |
| Recombinant DNA reagent | pLentiGuide-Puro (plasmid) | Other | | Gift from Dr. Mortiz Gaidt |
| Recombinant DNA reagent | Inflammasome reconstitution plasmids | PMID:27926929 | | Gifts from Dr. Russell Vance: human NLRP1 TEV (NLRP1-TEV2), human ASC, human and mouse CASP1, human IL-1B-V5, mouse IL-1B, and TEV protease |
| Recombinant DNA reagent | CVB3-Nancy infectious clone plasmid | PMID:2410905 | | Gift from Dr. Julie Pfeiffer |
| Recombinant DNA reagent | EMCV-Mengo infectious clone plasmid | PMID:2538661 | | Gift from Dr. Julie Pfeiffer |
| Commercial assay or kit | QUANTI-Blue assay reagent (for HEK-Blue IL-1β cells) | Invivogen | REP-QBS | Includes necessary reagents for measuring IL-1β release from HEK-Blue-IL1B reporter cell line |

*Continued on next page*

*Continued*

| Reagent type (species) or resource | Designation | Source or reference | Identifiers | Additional information |
|---|---|---|---|---|
| Chemical compound, drug | TransIT-X2 | Mirus | MIR 6000 | |
| Chemical compound, drug | MG132 | Sigma-Aldrich | M7449 | |
| Chemical compound, drug | MLN4924 | APExBIO | B1036 | |
| Antibody | V5-Tag rabbit monoclonal | Cell Signaling Technology | Cat# 13202; RRID:AB_2687461 | (1:1000) |
| Antibody | Flag-Tag mouse monoclonal | Sigma-Aldrich | Cat# F1804; RRID:AB_262044 | (1:2000) |
| Antibody | Myc-Tag rabbit monoclonal | Cell Signaling Technology | Cat# 2278; RRID:AB_490778 | (1:1000) |
| Antibody | HA-Tag rat monoclonal | Roche | Cat# 11867423001; RRID:AB_390918 | (1:1000) |
| Antibody | GAPDH rabbit monoclonal | Cell Signaling Technology | Cat# 2118; RRID:AB_561053 | (1:2000) |
| Antibody | Goat anti-Rat IgG (H+L) Secondary Antibody, HRP | Thermo Fisher Scientific | Cat# 31470; RRID:AB_228356 | (1:10000) |
| Antibody | Goat anti-Rabbit IgG (H+L) Secondary Antibody, HRP | Biorad | Cat# 170–6515; RRID:AB_11125142 | (1:10000) |
| Antibody | Goat anti-Mouse IgG (H+L) Secondary Antibody, HRP | Biorad | Cat# 170–6516; RRID:AB_11125547 | (1:10000) |
| Antibody | β-Tubulin mouse monoclonal | Sigma-Aldrich | Cat# T4026; RRID:AB_477577 | (1:2000) |
| Antibody | mouse IL-1β goat polyclonal | R and D Systems | Cat# AF-401-NA; RRID:AB_416684 | (1:1000) |
| Sequence-based reagent | Oligonucleotides | Other | | See *Supplementary file 5* for list of oligonucleotides used in this study |
| Software, algorithm | MEME v5.0.3 | PMID:25953851 | RRID:SCR_001783 | Motif finder (FIMO) |
| Software, algorithm | MAFFT 7.309 | PMID:23329690 | RRID:SCR_011811 | |
| Software, algorithm | NetSurfP | PMID:30785653 | RRID:SCR_018781 | http://www.cbs.dtu.dk/services/NetSurfP/ (Original); https://services.healthtech.dtu.dk/service.php?NetSurfP-2.0 (Alternate) |
| Software, algorithm | Geneious | PMID:22543367 | RRID:SCR_010519 | Neighbor-joining phylogenetic tree |
| Software, algorithm | BLASTp | PMID:9254694 | RRID:SCR_001010 | |

## Motif generation and search

To build the motif, 2658 nonredundant enteroviral polyprotein sequences were collected from the Viral Pathogen Resource (ViPR) and aligned with 20 well-annotated reference enteroviral polyprotein sequences from RefSeq (*Supplementary file 1*). P1 and P1' of the eight annotated cleavage sites across the RefSeq sequences served as reference points for putative cleavage sites across the 2658

ViPR sequences, with the exception of enterovirus D polyproteins. The 3C$^{pro}$ cleavage site for VP3-VP1 within polyproteins from the clade of enterovirus D have been described to be undetectable and have thus been removed (*Tan et al., 2013*). Four amino acyl residues upstream (P4-P1) and downstream (P1'-P4') of each cleavage site were extracted from every MAFFT-aligned polyprotein sequence, resulting in 2678 sets of cleavage sites (RefSeq sites included). Each set of cleavage sites representative of each polyprotein was then concatenated. Next, 1884 duplicates were removed from the 2678 concatenated cleavage sites. The remaining 796 nonredundant, concatenated cleavage sites were then split into individual 8-mer cleavage sites and the 6333 8-mers were aligned using MAFFT to generate Geneious-defined sequence logo information at each aligned position. Pseudo-counts to the position-specific scoring matrix were adjusted by total information content within each position relative to the two most information-dense position P1 and P1' (pseudocount = 0) and the least information-dense position P3 (pseudocount = 1). The 0.002 p-value threshold for FIMO motif searching against human NLRP1 was determined to optimize the capture of 95% of initial input cleavage sites within the set of 2678 whole enteroviral polyproteins and a majority sites within a previously described dataset of enteroviral 3C$^{pro}$ targets (*Laitinen et al., 2016*).

## NetSurfP

Prediction of the coil probability across human NLRP1 (NCBI accession NP_127497.1) was conducted using the protein FASTA as the input for the NetSurfP web server (http://www.cbs.dtu.dk/services/NetSurfP/).

## Sequence alignments, phylogenetic trees, and NLRP1 phylogenomics

Complete polyprotein sequences from 796 picornaviruses with non-redundant 3C$^{pro}$ cleavage sites (see 'Motif generation and search' section above) were downloaded from ViPR. Sequences were aligned using MAFFT (*Katoh and Standley, 2013*) and a neighbor-joining phylogenetic tree was generated using Geneious software (*Kearse et al., 2012*). An alignment and phylogenetic tree of all the 3C$^{pro}$ sequences used in this study was generated similarly.

To identify mammalian NLRP1 homologs, and species that lack NLRP1, the human NLRP1 protein sequence was used to query the RefSeq protein sequence database, a curated collection of the most well-assembled genomes, using BLASTp (*Altschul et al., 1997*). Sequences were downloaded and aligned using MAFFT implemented in Geneious software. Consensus sequence logos shown were generated using Geneious software. We determined that NLRP1 was 'absent' from a clade of species using the following criteria: (1) when searching with human NLRP1, we found an obvious homolog of another NLRP protein (generally NLRP3, NLRP12 or NLRP14) but no complete or partial homolog of NLRP1 and (2) this absence was apparent in every member of the clade of species (>2 species) in the RefSeq database.

## Plasmids and constructs

For NLRP1 cleavage assays, the coding sequences of human NLRP1 WT (NCBI accession NP_127497.1), human NLRP1 mutants (G131P, G131R, Q130R, G127E), human NLRP1 TEV or mouse NLRP1B (129 allele, NCBI accession AAZ40510.1; B6 allele, NCBI accession XM_017314698.2) were cloned into the pcDNA5/FRT/TO backbone (Invitrogen, Carlsbad, CA) with an N-terminal 3xFlag and mCherry tag. For NLRP1 activation, the same sequences were cloned into the pQCXIP vector backbone (Takara Bio, Mountain View, CA) with a C-terminal Myc tag (human NLRP1 sequences) or N-terminal EGFP and C-terminal HA (mouse NLRP1B sequences). Vectors containing the coding sequences of human NLRP1 TEV (NLRP1-TEV2), ASC, human and mouse CASP1, human IL-1β-V5, mouse IL-1β, and TEV protease (*Chavarría-Smith et al., 2016*) were generous gifts from Dr. Russell Vance, UC Berkeley. Single point mutations were made using overlapping stitch PCR. A list of primers used to generate the wild-type and mutant NLRP1 constructs are described in *Supplementary file 5*.

CVB3 3C$^{pro}$ and EMCV 3C$^{pro}$ were cloned from CVB3-Nancy and EMCV-Mengo plasmids (generous gifts from Dr. Julie Pfeiffer, UT Southwestern). Remaining 3C$^{pro}$ sequences were ordered as gBlocks (Integrated DNA Technologies, San Diego, CA). Each 3C$^{pro}$ was cloned with an N-terminal HA tag into the QCXIP vector backbone. Catalytic mutations were made using overlapping stitch

PCR. A list of primers and gBlocks used to generate the protease constructs are described in *Supplementary file 5*.

Following cloning, all plasmid stocks were sequenced across the entire inserted region to verify that no mutations were introduced during the cloning process.

## Cell culture and transient transfection

All cell lines (HEK293T, HEK-Blue-IL-1β, and HaCaT) are routinely tested for mycoplasma by PCR kit (ATCC, Manassas, VA) and kept a low passage number to maintain less than one year since purchase, acquisition or generation. HEK293T cells were obtained from ATCC (catalog # CRL-3216), HEK-Blue-IL-1β cells were obtained from Invivogen (catalog # hkb-il1b) and HaCaT cells were obtained from the UC Berkeley Cell Culture Facility (https://bds.berkeley.edu/facilities/cell-culture) and all lines were verified by those sources. All cells were grown in complete media containing DMEM (Gibco, Carlsbad, CA), 10% FBS (Peak Serum, Wellington, CO), and appropriate antibiotics (Gibco, Carlsbad, CA). For transient transfections, HEK293T cells were seeded the day prior to transfection in a 24-well plate (Genesee, El Cajon, CA) with 500 µl complete media. Cells were transiently transfected with 500 ng of total DNA and 1.5 µl of Transit X2 (Mirus Bio, Madison, WI) following the manufacturer's protocol. HEK-Blue IL-1β reporter cells (Invivogen, San Diego, CA) were grown and assayed in 96-well plates (Genesee, El Cajon, CA).

## HaCaT knockouts

To establish *NLRP1* and *CASP1* knockouts in human immortalized keratinocyte HaCaT cells, lentivirus-like particles were made by transfecting HEK293T cells with the plasmids psPAX2 (gift from Didier Trono, Addgene plasmid # 12260) and pMD2.G (gift from Didier Trono, Addgene plasmid # 12259) and either the pLB-Cas9 (gift from Feng Zhang, Addgene plasmid # 52962) (*Sanjana et al., 2014*) or a plentiGuide-Puro, which was adapted for ligation-independent cloning (kindly gifted by Moritz Gaidt) (*Schmidt et al., 2015*). Guide sequences are shown in *Supplementary file 5*. Conditioned supernatant was harvested 48 and 72 hr post-transfection and used for spinfection of HaCaT cells at 1200 x *g* for 90 min at 32°C. Forty-eight hours post-transduction, cells with stable expression of Cas9 were selected in media containing 100 µg/ml blasticidin. Blasticidin-resistant cells were then transduced with sgRNA-encoding lentivirus-like particles, and selected in media containing 1.3 µg/ml puromycin. Cells resistant to blasticidin and puromycin were single cell cloned by limiting dilution in 96-well plates, and confirmed as knockouts by Sanger sequencing (*Figure 3—figure supplement 3*).

## NLRP1 cleavage assays

100 ng of epitope-tagged human NLRP1 WT, human NLRP1 mutants (G131P, G131R, Q130R, G127E), human NLRP1 TEV or mouse NLRP1B was co-transfected with 250 ng of HA-tagged protease-producing constructs. Twenty-four hours post-transfection, the cells were harvested, lysed in 1x NuPAGE LDS sample buffer (Invitrogen, Carlsbad, CA) containing 5% β-mercaptoethanol (Fisher Scientific, Pittsburg, PA) and immunoblotted with antibodies described below.

## NLRP1 activity assays

For human NLRP1 activation assays, 5 ng of ASC, 100 ng of CASP1, 50 ng of IL-1β-V5, and 100 ng of various protease-producing constructs were co-transfected with 4 ng of either pQCXIP empty vector, wild-type or mutant pQCXIP-NLRP1-Myc constructs. For inhibitor treatments, cells were treated with either 0.5 µM MG132 or 1.0 µM MLN4924 18 hr after transfection. Twenty-four hours post-transfection, cells were harvested and lysed in 1x NuPAGE LDS sample buffer containing 5% β-mercaptoethanol or in 1x RIPA lysis buffer with protease inhibitor cocktail (Roche) and immunoblotted with antibodies described below or culture media was harvested for quantification of IL-1β levels by HEK-Blue assays (see below).

For mouse NLRP1B activation assays, 50 ng of mouse CASP1, 50 ng of mouse IL-1β, and 100 ng of various protease-producing constructs were co-transfected with either 4 ng of 129 NLRP1B or 2.5 ng B6 NLRP1B constructs. Twenty-four hours post-transfection, cells were harvested in 1x RIPA lysis buffer with protease inhibitor cocktail (Roche) and immunoblotted with antibodies described below.

## Viral stocks and viral infections

CVB3 and EMCV viral stocks were generated by co-transfection of CVB3-Nancy or EMCV-Mengo infectious clone plasmids with a plasmid expressing T7 RNA polymerase (generous gifts from Dr. Julie Pfeiffer, UT Southwestern) as previously described (*McCune et al., 2020*). Supernatant was harvested, quantified by plaque assay or TCID50 on HEK293T cells, and frozen in aliquots at −80°C.

For viral infections of HEK293T cells, cells were transfected in 24-well plates and infected with 250,000 PFU (MOI = ~1) CVB3 or EMCV or mock infected for the indicated times. For cleavage assays, cells were transfected with 100 ng of the indicated NLRP1 construct and, 24 hr after transfection, cells were harvested and lysed in 1x NuPAGE LDS sample buffer containing 5% β-mercaptoethanol and immunoblotted with antibodies described below. For activation assays, cells were transfected with 4 ng of the indicated NLRP1 construct and 5 ng of ASC, 100 ng of CASP1, 50 ng of IL-1β-V5. Twenty-four hours after transfections, cells were infected with virus (or mock infected) and culture supernatant was collected 8 hr later (32 hr post-transfection). Culture supernatant was filtered through a 100,000 MWCO centrifugal spin filter (MilliporeSigma, Burlington, MA) for 10 min at 12,000xg to remove infectious virus and IL-1β levels were quantified by HEK-Blue assays (see below).

For viral infections of HaCaT cells, cells were plated in 24-well plates. The next day, cells were infected with 100,000 PFU/well (MOI = ~0.4) CVB3 or EMCV or mock infected. Forty-eight hours after infection, culture supernatant was collected, spin filtered as described above to remove infectious virus, and IL-1β levels were quantified by HEK-Blue assays (see below).

## HEK-Blue IL-1β assay

To quantify the levels of bioactive IL-1β released from cells, we employed HEK-Blue IL-1β reporter cells (Invivogen, San Diego, CA). In these cells, binding to IL-1β to the surface receptor IL-1R1 results in the downstream activation of NF-kB and subsequent production of secreted embryonic alkaline phosphatase (SEAP) in a dose-dependent manner (*Figure 3—figure supplement 1*). SEAP levels are detected using a colorimetric substrate assay, QUANTI-Blue (Invivogen, San Diego, CA) by measuring an increase in absorbance at $OD_{655}$.

Culture supernatant from inflammasome-reconstituted HEK293T cells or HaCaT cells that had been transfected with $3C^{pro}$ or virally infected (see above) was added to HEK-Blue IL-1β reporter cells plated in 96-well format in a total volume of 200 µl per well. On the same plate, serial dilutions of recombinant human IL-1β (Invivogen, San Diego, CA) were added in order to generate a standard curve for each assay. Twenty-four hours later, SEAP levels were assayed by taking 20 µl of the supernatant from HEK-Blue IL-1β reporter cells and adding to 180 µl of QUANTI-Blue colorimetric substrate following the manufacturer's protocol. After incubation at 37°C for 30–60 min, absorbance at $OD_{655}$ was measured on a BioTek Cytation five plate reader (BioTek Instruments, Winooski, VT) and absolute levels of IL-1β were calculated relative to the standard curve. All assays, beginning with independent transfections or infections, were performed in triplicate.

## Immunoblotting and antibodies

Harvested cell pellets were washed with 1X PBS, and lysed with 1x NuPAGE LDS sample buffer containing 5% β-mercaptoethanol at 98C for 10 min. The lysed samples were spun down at 15000 RPM for two minutes, followed by loading into a 4–12% Bis-Tris SDS-PAGE gel (Life Technologies, San Diego, CA) with 1X MOPS buffer (Life Technologies, San Diego, CA) and wet transfer onto a nitrocellulose membrane (Life Technologies, San Diego, CA). Membranes were blocked with PBS-T containing 5% bovine serum albumin (BSA) (Spectrum, New Brunswick, NJ), followed by incubation with primary antibodies for V5 (IL-1β), FLAG (mCherry-fused NLRP1 for protease assays), Myc (NLRP1-Myc for activation assays), HA (viral protease or mouse NLRP1B), β-tubulin, or GAPDH. Membranes were rinsed three times in PBS-T then incubated with the appropriate HRP-conjugated secondary antibodies. Membranes were rinsed again three times in PBS-T and developed with SuperSignal West Pico PLUS Chemiluminescent Substrate (Thermo Fisher Scientific, Carlsbad, CA). The specifications, source, and clone info for antibodies are described in *Supplementary file 6*.

## Acknowledgements

We thank members of the Daugherty laboratory and members of the San Diego Program in Immunology for helpful suggestions, Julie Pfeiffer for reagents and protocols for enterovirus virus generation, and Russell Vance, Andrew Sandstrom, and members of the Daugherty laboratory for critical reading of the manuscript. This work was supported by the National Institutes of Health (R35 GM133633), Pew Biomedical Scholars Program and Hellman Fellows Program to MDD, T32 GM007240 to BVT, CB and APR, a National Science Foundation graduate research fellowship (2019284620) to CB, and a Jane Coffin Childs Memorial Fund for Medical Research postdoctoral fellowship to PSM.

## Additional information

### Funding

| Funder | Grant reference number | Author |
|---|---|---|
| National Institutes of Health | R35 GM133633 | Matthew D Daugherty |
| Pew Charitable Trusts | | Matthew D Daugherty |
| Hellman Foundation | | Matthew D Daugherty |
| National Institutes of Health | T32 GM007240 | Brian V Tsu<br>Christopher Beierschmitt<br>Andrew P Ryan |
| National Science Foundation | 2019284620 | Christopher Beierschmitt |
| Jane Coffin Childs Memorial Fund for Medical Research | | Patrick S Mitchell |

The funders had no role in study design, data collection and interpretation, or the decision to submit the work for publication.

### Author contributions

Brian V Tsu, Conceptualization, Data curation, Software, Formal analysis, Validation, Investigation, Visualization, Methodology, Writing - original draft, Writing - review and editing; Christopher Beierschmitt, Formal analysis, Validation, Investigation, Visualization, Methodology, Writing - review and editing; Andrew P Ryan, Resources, Investigation, Methodology, Writing - review and editing; Rimjhim Agarwal, Investigation, Writing - review and editing; Patrick S Mitchell, Conceptualization, Resources, Investigation, Methodology, Writing - review and editing; Matthew D Daugherty, Conceptualization, Data curation, Formal analysis, Supervision, Funding acquisition, Validation, Investigation, Visualization, Methodology, Writing - original draft, Project administration, Writing - review and editing

### Author ORCIDs

Brian V Tsu  https://orcid.org/0000-0003-0268-8323
Christopher Beierschmitt  https://orcid.org/0000-0003-0151-1091
Andrew P Ryan  https://orcid.org/0000-0002-2630-9837
Patrick S Mitchell  https://orcid.org/0000-0001-8375-9060
Matthew D Daugherty  https://orcid.org/0000-0002-4879-9603

### Decision letter and Author response

Decision letter https://doi.org/10.7554/eLife.60609.sa1
Author response https://doi.org/10.7554/eLife.60609.sa2

# Additional files

## Supplementary files

• Supplementary file 1. Training set of enteroviral polyproteins. Accession IDs are listed for all poly-proteins used to benchmark the motif search described in *Figure 1D* and *Figure 1—figure supplement 1*. The 8mer cleavage sites and concatenated 8mer sequences are included.

• Supplementary file 2. Enteroviral polyproteins with unique 8mer 3C$^{pro}$ cleavage site concatenations. Accession IDs are listed for all polyproteins used to create the search motif shown in *Figure 1C* and *Figure 1—figure supplement 1* and the enteroviral phylogenetic tree in *Figure 1B*. The 8mer cleavage sites and concatenated 8mer sequences are included.

• Supplementary file 3. Un-optimized 3C$^{pro}$ cleavage motif scores for true positive, false positive and human sites within the enteroviral polyprotein and human training sets. FIMO-generated p-values and $\log_{10}$(p-value) represent the cleavage score at the provided matched sequence, where (A) is the un-optimized 3C$^{pro}$ cleavage motif scores for true positive hits within enteroviral polyprotein dataset, (B) is the un-optimized 3C$^{pro}$ cleavage motif scores for false positive hits within enteroviral polyprotein dataset where unique site matches are shown (26062), and (C) is the un-optimized 3C$^{pro}$ cleavage motif scores for reported human cleavage sites from the *Laitinen et al., 2016* dataset.

• Supplementary file 4. Optimized 3C$^{pro}$ cleavage motif scores for true positive, false positive and human sites within the enteroviral polyprotein and human training sets. FIMO-generated p-values and $\log_{10}$(p-value) represent the cleavage score at the provided matched sequence, where (A) is the optimized 3C$^{pro}$ cleavage motif scores for true positive hits within enteroviral polyprotein dataset, (B) is the optimized 3C$^{pro}$ cleavage motif scores for false positive hits within enteroviral polyprotein dataset where unique site matches are shown (24437), and (C) is the optimized 3C$^{pro}$ cleavage motif scores for reported human cleavage sites from the *Laitinen et al., 2016* dataset.

• Supplementary file 5. List of primers and gBlocks used. Names and notes contain details on the restriction enzyme sites or point mutations encoded.

• Supplementary file 6. List of antibodies used for immunoblots. Manufacturer and dilutions used are noted.

• Supplementary file 7. List of accession numbers used for sequence alignments.

• Transparent reporting form

## Data availability

All data generated or analyzed during this study are included in the manuscript and supporting files. Sources of sequence information used for figures and figure supplements have been provided. The ViPR database was used to collect enteroviral polyprotein sequences using the Picornaviridae-specific Gene/Protein search tool (https://www.viprbrc.org/brc/vipr_protein_search.spg?method=Show-CleanSearch&decorator=picorna), selecting protein sequences from all enteroviruses with filters for complete genome to include "completely genome only" and a search type to "include Polyproteins in Results" with the Gene Product Name of "polyprotein". Using the advanced options, options for a minimum CDS length of "6000" with "remove duplicate sequences" were selected. The collection of sequences used in this analysis are listed in Supplementary files 1 and 2. The NCBI protein database (https://www.ncbi.nlm.nih.gov/protein) was used to collect sequences for human (NP_127497.1), mouse NLRP1B allele 129 (AAZ40510.1), mouse NLRP1B allele B6 (XM_017314698.2), other mammalian NLRP1 sequences (Supplementary file 7), picornaviral 3C protease sequences (Supplementary file 7), and NCBI RefSeq enterovirus polyprotein sequences. The NCBI RefSeq enterovirus polyprotein sequences were collected from the NCBI protein database using the search phrase "Enterovirus[Organism] AND srcdb_refseq[PROP] NOT cellular organisms[ORGN]" and filtering by sequence length "2000 to 4000" and release date "to 2018/04/31". Human non-synonymous allele counts for NLRP1 (Figure 4C) were collected using gnomAD (https://gnomad.broadinstitute.org/) v2.1.1 with the search term "NLRP1".

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
