## [Decision Letter]

**Acceptance summary:**

Bacterial proteases activate the NLRP1B inflammasome in mice, but pathogen-derived activators of the human NLRP1 inflammasome are less well characterized. Here, the authors demonstrate that certain picornaviruses, via their 3C protease, trigger host immune responses by activating human NLRP1, thereby revealing a previously unappreciated sensing mechanism for virally-encoded proteases. Genetic approaches suggest this mechanism has adapted throughout primate evolution as a way to give the host an advantage with respect to its ongoing genetic conflict with the virus.

**Decision letter after peer review:**

Thank you for submitting your article "Diverse viral proteases activate the NLRP1 inflammasome" for consideration by *eLife*. Your article has been reviewed by three peer reviewers, one of whom is a member of our Board of Reviewing Editors, and the evaluation has been overseen by Carla Rothlin as the Senior Editor. The reviewers have opted to remain anonymous.

The reviewers have discussed the reviews with one another and the Reviewing Editor has drafted this decision to help you prepare a revised submission.

Summary:

The anthrax lethal factor protease activates the NLRP1B inflammasome in mice, but pathogen-derived activators of the human NLRP1 inflammasome have not yet been identified. Here, the authors demonstrate that expression of picornavirus 3C protease activates human NLRP1, implicating a sensing mechanism for pathogen-encoded proteases. Computational and functional evolutionary approaches suggest this mechanism has adapted in primates in host-specific and virus-specific manners.

Essential revisions:

The reviewers were uniform in their requirement for demonstration of this mechanism during picornavirus infection, as all of the current studies are with ectopically expressed protease. Additional experiments to solidify conclusions are outlined below.

1a) Reviewer #1: Does 3C from an active viral infection cleave NLRP1 in the transfection system? The authors could pick 1 enterovirus and 1 non-enterovirus from Figure 4B and infect cells transfected with the NLRP1 system, then score cleavage, Ilb secretion etc.

1b) Reviewer #2: The biggest weakness with this study is that the authors only showed that ectopically expressed proteases can cleave human NLRP1. It is not clear from this work if a living virus will actually cleave and activate NLRP1. If the authors could confirm that the virus itself cleaves and activates NLRP1, it would significantly increase the impact of this work.

1c) Reviewer #3: The manuscript is lacking the biological relevance that would come from carrying out the cleavage experiments in cells infected by coxsackievirus B3, or one of the other enteroviruses whose 3C proteases cleave NLRP1 in transfected cells. Unlike the transfection assays reported in the manuscript, an infection delivers the viral genomic RNA into the cytoplasm of infected cells where translation of the genome produces a polyprotein that is then processed by the 3C and 2A protease activities to generate the viral proteins. In the transfection assays described in this manuscript, the plasmids are delivered to the nucleus, where mRNAs are generated and subsequently transported to the cytoplasm where they are translated as capped, monocistronic mRNAs that produce a single mature 3C protease. These major biological differences could produce differing cleavage patterns of NLRP1 based upon cytoplasmic location of viral proteins or differing forms of the different precursors containing viral protease sequences, which may have differing accessibilities to host protein cleavage targets. Also, the levels of 3C protease produced in cells transfected with plasmid expression vectors may be quite different than what is produced during a coxsackievirus B3 infection. Since the levels of 3C protease produced in the cytoplasm would drive the extent of NRPL1 cleavage and inflammasome activation, the authors need to carry out their assays in the context of a bona fide viral infection.

1d) Reviewer #3: For the related enteroviruses being used in this study, the mature 3C protease is not the major source of protease produced during infection by these viruses. Rather it is the 3CD polypeptide precursor that carries out the majority of cleavage events due to its abundance during an infection. Thus, the use of a mature 3C protease in the transfection assays does not mimic an authentic viral infection in terms of which 3C containing enzymes might be capable of cleaving NLRP1 (or not).

2) As there is little if any information on the role of NLRP1 in human enterovirus infection, the authors should demonstrate that enterovirus infection cleaves NLRP1 in relevant primary cell type such as human monocyte-derived macrophages, or primary human dermal fibroblasts or keratinocytes. A comparison to a picornavirus that does not cleave NLRP1 would be most impactful.

3) A nice control would be to show that a proteasome inhibitor blocks inflammasome activation by 3C^pro^.

4) In Figure 4D, the authors show that some proteases cleave mouse NLRP1B, but it is not demonstrated if that activates the inflammasome. An IL-1b ELISA or LDH assay should be shown as well.

---

## [Author Response]

Essential revisions:The reviewers were uniform in their requirement for demonstration of this mechanism during picornavirus infection, as all of the current studies are with ectopically expressed protease. Additional experiments to solidify conclusions are outlined below.1a) Reviewer #1: Does 3C from an active viral infection cleave NLRP1 in the transfection system? The authors could pick 1 enterovirus and 1 non-enterovirus from Figure 4B and infect cells transfected with the NLRP1 system, then score cleavage, Ilb secretion etc.1b) Reviewer #2: The biggest weakness with this study is that the authors only showed that ectopically expressed proteases can cleave human NLRP1. It is not clear from this work if a living virus will actually cleave and activate NLRP1. If the authors could confirm that the virus itself cleaves and activates NLRP1, it would significantly increase the impact of this work.1c) Reviewer #3: The manuscript is lacking the biological relevance that would come from carrying out the cleavage experiments in cells infected by coxsackievirus B3, or one of the other enteroviruses whose 3C proteases cleave NLRP1 in transfected cells. Unlike the transfection assays reported in the manuscript, an infection delivers the viral genomic RNA into the cytoplasm of infected cells where translation of the genome produces a polyprotein that is then processed by the 3C and 2A protease activities to generate the viral proteins. In the transfection assays described in this manuscript, the plasmids are delivered to the nucleus, where mRNAs are generated and subsequently transported to the cytoplasm where they are translated as capped, monocistronic mRNAs that produce a single mature 3C protease. These major biological differences could produce differing cleavage patterns of NLRP1 based upon cytoplasmic location of viral proteins or differing forms of the different precursors containing viral protease sequences, which may have differing accessibilities to host protein cleavage targets. Also, the levels of 3C protease produced in cells transfected with plasmid expression vectors may be quite different than what is produced during a coxsackievirus B3 infection. Since the levels of 3C protease produced in the cytoplasm would drive the extent of NRPL1 cleavage and inflammasome activation, the authors need to carry out their assays in the context of a bona fide viral infection.

We have now included three additional sets of experiments that use live virus infections to address the reviewers’ concerns described above. All of these experiments have been done in parallel with two related picornaviruses that we predict should have different abilities to cleave and activate human NLRP1 based on our data with transfected 3C^pros^ from these viruses. Specifically, based on the activity of the ectopically-expressed proteases, we predicted that infection with CVB3 should result in cleavage and activation of human NLRP1 whereas infection with a related picornavirus, EMCV, should result in neither cleavage nor activation of human NLRP1.

First, in Figure 2E, we show that infection with CVB3 results in cleavage of human NLRP1 that is detectable starting six hours after infection. Cleavage occurs at the same site (residues 130/131) that we observe with transfected protease. Importantly, in Figure 5D, we show that infection by EMCV results in no NLRP1 cleavage. These results from viral infections mirror the 3C^pro^ cleavage data found in Figure 5B.

Second, in Figure 3C and Figure 5—figure supplement 5, we show that infection with CVB3, but not EMCV, can activate the human NLRP1 inflammasome in transfected HEK293T cells. Virus-mediated NLRP1 activation requires cleavage at site 130/131, recapitulating the viral and host specificity we observe in Figure 5C.

Finally, we identified an immortalized human keratinocyte cell line, HaCaT, to test whether viral infection could activate endogenously expressed NLRP1. We first demonstrated that infection of this cell line with CVB3, but not EMCV, results in IL-1b release. We went on to generate CRISPR knockouts of NLRP1 and CASP1 and showed reduced IL-1b release when these cells were infected with CVB3. These data are now shown in Figure 3D and Figure 5—figure supplement 5.

Together, these data show that viral infection recapitulates the molecular specificity that we describe with ectopically expressed viral proteases, and also demonstrate that viral infection of human keratinocytes results in NLRP1-dependent inflammasome activation.

Also, as we cite in our paper, while we were working on generating these additional experimental data, another paper was published (Robinson et al., 2020) that convergently discovered that 3C^pro^ can cleave and activate NLRP1. In their paper, they focus primarily on human rhinovirus (HRV) 3C^pro^ and show, using approaches similar to the ones we took, that infection of airway epithelia with HRV results in NLRP1 inflammasome activation. The concordance of these data with ours highlight the relevance of 3C^pro^-mediated cleavage and activation of NLRP1 during picornaviral infection.

1d) Reviewer #3: For the related enteroviruses being used in this study, the mature 3C protease is not the major source of protease produced during infection by these viruses. Rather it is the 3CD polypeptide precursor that carries out the majority of cleavage events due to its abundance during an infection. Thus, the use of a mature 3C protease in the transfection assays does not mimic an authentic viral infection in terms of which 3C containing enzymes might be capable of cleaving NLRP1 (or not).

As described above, we now include viral infection data that indicates that the protease generated during viral infection cleaves and activates NLRP1 with the same specificity that we observe with ectopically expressed 3C^pros^. Although we do not formally know whether this activity is performed by mature 3C^pro^ or by 3CD, we have also made changes in the Results section that acknowledge the role that 3CD plays as a protease during viral infection.

2) As there is little if any information on the role of NLRP1 in human enterovirus infection, the authors should demonstrate that enterovirus infection cleaves NLRP1 in relevant primary cell type such as human monocyte-derived macrophages, or primary human dermal fibroblasts or keratinocytes. A comparison to a picornavirus that does not cleave NLRP1 would be most impactful.

As described above, we have now shown that CVB3 infection of immortalized human keratinocytes (HaCaT cell line) results in NLRP1 inflammasome activation. As per the reviewers’ suggestion, we also tested EMCV and showed that the NLRP1 inflammasome is not activated. Our CVB3 results are consistent with Robinson et al., who show that HRV infection of primary airway epithelia can activate the NLRP1 inflammasome.

3) A nice control would be to show that a proteasome inhibitor blocks inflammasome activation by 3C^pro^.

We have now included data in Figure 3—figure supplement 1 that shows three additional controls that support that the 3C^pro^-mediated NLRP1 activation we observe is consistent with the previously described mechanism of NLRP1 activation. First, we show that 3C^pro^ activation of NLRP1 requires auto-processing of the FIIND domain. We also show that treatment with either a chemical inhibitor of the proteasome (MG132) or a chemical inhibitor of the Cullin-RING E3 ubiquitin ligases that are required for the degradation of proteins with a novel N-terminal glycine (MLN4924) reduce the amount of NLRP1 inflammasome activation we see in response to 3C^pro^ cleavage.

4) In Figure 4D, the authors show that some proteases cleave mouse NLRP1B, but it is not demonstrated if that activates the inflammasome. An IL-1b ELISA or LDH assay should be shown as well.

We were also intrigued by this observation and chose to pursue the cleavage and activation of mouse NLRP1B further. We have included a new Figure 6 that compares cleavage of the 129 allele of NLRP1B with the B6 allele of NLRP1B by a panel of enterovirus 3C^pros^. At the reviewer’s request, we have also included parallel activation data in Figure 6B and show, in particular, that cleavage of B6 NLRP1B by rhinovirus 3C^pro^ results in strong inflammasome activation as measured by IL-1β maturation. Although NLRP1B is not the main focus of the paper, these data add to the overall model that host and virus evolution, even between closely related species or strains, impact 3C^pro^ activation of NLRP1. We also note that rhinovirus 3C^pro^ is the first described virus-encoded activator of mouse NLRP1B, and that activation only occurs for B6 NLRP1B. We are therefore optimistic that these observations could be a good starting point for studies that aim to leverage mouse infection models to reveal the in vivo antiviral and immunopathological consequences of viral-mediated NLRP1 inflammasome activation.